# LADA: Scalable Label-Specific CLIP Adapter for Continual Learning

**Mao-Lin Luo** [1 2]   **Zi-Hao Zhou** [1 2]   **Tong Wei** [1 2]   **Min-Ling Zhang** [1 2]

## Abstract

Continual learning with vision-language models like CLIP offers a pathway toward scalable machine learning systems by leveraging its transferable representations. Existing CLIP-based methods adapt the pre-trained image encoder by adding multiple sets of learnable parameters, with each task using a partial set of parameters. This requires selecting the expected parameters for input images during inference, which is prone to error that degrades performance. To address this problem, we introduce LADA (**L**abel-specific **ADA**pter). Instead of partitioning parameters across tasks, LADA appends lightweight, label-specific memory units to the frozen CLIP image encoder, enabling discriminative feature generation by aggregating task-agnostic knowledge. To prevent catastrophic forgetting, LADA employs feature distillation for seen classes, preventing their features from being interfered with by new classes. Positioned after the image encoder, LADA prevents gradient flow to the frozen CLIP parameters, ensuring efficient training. Extensive results show that LADA achieves state-of-the-art performance in continual learning settings. The implementation code is available at https://github.com/MaolinLuo/LADA.

## 1. Introduction

Pre-trained vision-language models, such as CLIP (Radford et al., 2021), have become natural continual learners due to their ability to transfer representations across diverse tasks. Recently, several fine-tuning approaches for CLIP have been proposed to improve performance in downstream tasks, including full parameter fine-tuning and parameter-efficient fine-tuning (Gao et al., 2024; Wortsman et al., 2022a; Zhang et al., 2022; Wei et al., 2024; Gan & Wei, 2024; Shi et al., 2024). However, adapting CLIP representations using only task-specific data can severely impair the "general" knowledge encoded in the pre-trained CLIP parameters (Luo et al., 2023). This challenge is particularly problematic in continual learning settings, where the model must not only learn incrementally from a series of tasks, but also retain its previously acquired knowledge. As the model adapts to new tasks, the performance of previously learned knowledge often declines, a phenomenon known as *catastrophic forgetting* (French, 1999; McCloskey & Cohen, 1989). This issue highlights the trade-off between **memory stability** and **learning plasticity**: too much focus on the former can interfere with the latter, and vice versa (Wang et al., 2024b).

*Stability* in continual learning is closely related to catastrophic forgetting and can be divided into two aspects: forgetting newly learned tasks and forgetting pre-trained general knowledge. This dual forgetting phenomenon is characterized by *forward forgetting*, where the general knowledge in the pre-trained model degrades when making predictions on unseen tasks, and *backward forgetting*, where previously learned knowledge of seen tasks is lost (Tang et al., 2024). Although backward forgetting has been effectively mitigated through techniques such as regularization (Ahn et al., 2019; Zenke et al., 2017), prototype augmentation (Zhu et al., 2021; McDonnell et al., 2023; Zhang et al., 2023; Huang et al., 2024), and replay (Rolnick et al., 2019; Rebuffi et al., 2017), addressing forward forgetting remains a significant challenge due to the unavailability of pre-training data. To mitigate forward forgetting, ZSCL (Zheng et al., 2023) distills knowledge from a vanilla zero-shot CLIP model as a teacher into a fine-tuned CLIP model, using regularization to reduce forward forgetting. However, this approach still struggles to preserve the pre-trained knowledge due to potential updates to pre-trained parameters. Furthermore, many methods (Yu et al., 2024; Zheng et al., 2023) rely on external reference datasets, such as ImageNet and Conceptual Captions, to maintain generalization on general tasks. However, this is often impractical in real-world scenarios (Tang et al., 2024; Xu et al., 2024).

For *plasticity*, full-parameter tuning methods, such as ZSCL (Zheng et al., 2023), distill features from vanilla CLIP to mitigate catastrophic forgetting. However, such

[1]School of Computer Science and Engineering, Southeast University, Nanjing 210096, China [2]Key Laboratory of Computer Network and Information Integration (Southeast University), Ministry of Education, China. Correspondence to: Tong Wei <weit@seu.edu.cn>.

*Proceedings of the 42$^{nd}$ International Conference on Machine Learning*, Vancouver, Canada. PMLR 267, 2025. Copyright 2025 by the author(s).

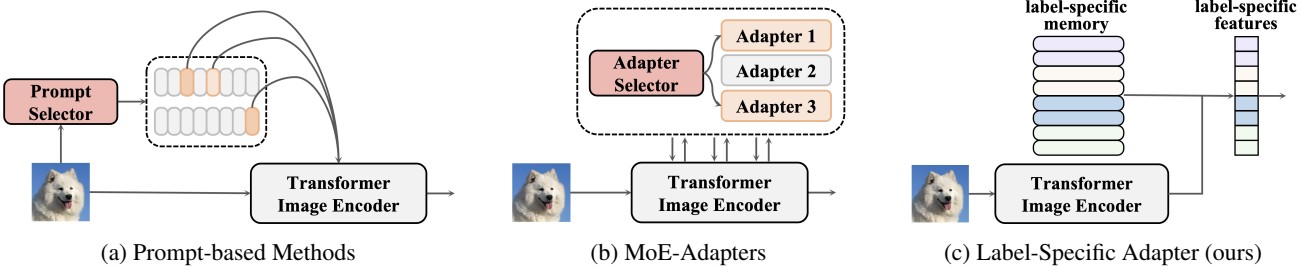

*Figure 1.* Comparison of CLIP tuning paradigms in continual learning. Our label-specific adapter leverages learned memory of all seen tasks and CLIP representations to generate label-specific features, eliminating the need for parameter selection.

approaches can impede the model's ability to effectively learn new tasks, leading to a suboptimal trade-off. Other prompt-based methods, such as L2P (Wang et al., 2022c), DualPrompt (Wang et al., 2022b) and S-Prompts (Wang et al., 2022a) expand the prompt pool as new tasks are learned, but their performance is limited by the constrained length of prompt tokens (Zheng et al., 2023; Tang et al., 2024). MoE-Adapters (Yu et al., 2024) inserts mixture of adapters into the image encoder to adapt CLIP representations by activating a few adapters for each task. Although effective, the number of adapters must be predefined, requiring prior knowledge of the total number of tasks. More importantly, since only a partial set of parameters is activated during training for each task, both prompt-based methods and MoE-Adapters involve an auxiliary parameter selection step to extract image features using the expected prompts or adapters (as illustrated in Figure 1). This selection process can lead to misassignments, significantly reducing classification performance.

To address these problems, we introduce a novel label-specific CLIP adapter, LADA, which appends a compact set of learnable label-specific vectors after the CLIP image encoder. LADA handles new tasks by sequentially adding a few label-specific vectors for each class, which act as cached memory units for seen classes. These vectors generate label-specific features by calculating their inner product with the CLIP representation. Memory units are learned by encouraging higher activations of the calculated label-specific features at positions corresponding to the ground-truth label. LADA freezes the label-specific memory units of previous tasks and only updates the memory for the new task, using both new task samples and the distilled features of previous tasks, reducing the risk of catastrophic forgetting. The design of LADA condenses discriminative information from all tasks, eliminating the need for parameter selection. Moreover, LADA is efficient for training as it does not require gradient propagation to the CLIP image encoder. In summary, **our contributions** are as follows:

- We introduce LADA, a lightweight CLIP adapter that condenses task-agnostic knowledge to transform CLIP

representation into label-specific features, eliminating the need for parameter selection as seen in previous mainstream methods.

- Our method is scalable and efficient, adding only small sets of learnable memory units for novel tasks, without requiring gradient propagation to the frozen CLIP image encoder.

- We achieve state-of-the-art results in both 16-shot and full-shot continual learning settings, surpassing previous methods in *Transfer*, *Average*, and *Last* accuracy.

## 2. Related Works

**Continual Learning Settings.** Early continual learning research mainly considers Task-Incremental Learning (TIL) (Hsu, 2018), where models sequentially learn tasks and rely on task identities at inference time. Later, Class-Incremental Learning (CIL) (Rebuffi et al., 2017) removes this reliance, compelling models to distinguish among all encountered classes without explicit task cues. Both TIL and CIL generally assume that all tasks originate from a single dataset (Yan et al., 2021; Zhu et al., 2021; Douillard et al., 2022), thus confining themselves to a single-domain scenario. In contrast, Multi-domain Task-Incremental Learning (MTIL) (Zheng et al., 2023) introduces tasks drawn from diverse sources that each demand specialized domain knowledge from animal species to aircraft series, reflecting the complexity of real-world applications. Recently, Cross-domain Task-Agnostic Incremental Learning (X-TAIL) (Xu et al., 2024) has taken one step further by discarding task or domain identity, pushing the setting closer to practical scenarios.

**Continual Learning Methods.** Prevailing continual learning methods can be broadly classified into three categories, including replay-based, regularization-based and architecture-based approaches based on surveys (De Lange et al., 2021; Wang et al., 2024b; Roth et al., 2024). *Replay-based methods* (Rebuffi et al., 2017; Isele & Cosgun, 2018; Lavda et al., 2018; Lopez-Paz & Ranzato, 2017; Shin et al., 2017; Rolnick et al., 2019) store a subset of previous task data, which is replayed alongside new task data during train-

ing to mitigate catastrophic forgetting. These methods, such as Experience Replay (Isele & Cosgun, 2018; Rolnick et al., 2019) and Generative Replay (Lavda et al., 2018; Shin et al., 2017), aim to maintain the performance on prior tasks by revisiting old examples or generating pseudo-samples using a generative model. While effective, the need to store or generate exemplars raises concerns regarding memory efficiency and scalability. *Regularization-based methods* (Zheng et al., 2023; Ahn et al., 2019; Kirkpatrick et al., 2017; Jha et al., 2024; Zenke et al., 2017) introduce additional regularization terms in the loss function to mitigate catastrophic forgetting, balancing between old and new tasks. One such method, ZSCL (Zheng et al., 2023), regularizes the model parameters by penalizing shifts in the parameter space, thus preserving the robustness of pre-trained models even without direct access to the original training data. *Architecture-based methods* (Wang et al., 2024a; Lu et al., 2024; Li et al., 2019; Houlsby et al., 2019; Yoon et al., 2018) enhance continual learning by expanding the model architecture, often by adding new parameters or components dedicated to specific tasks. Prompt-based methods, such as L2P (Wang et al., 2022c) and DualPrompt (Wang et al., 2022b), expand the prompt pool as new tasks are learned, relying on auxiliary losses. CODA-Prompt (Smith et al., 2023) proposes end-to-end prompt selection methods to increase plasticity. APG (Tang et al., 2023) introduces a prompt generator to reduce the gap between pretraining and future tasks, while EvoPrompt (Kurniawan et al., 2024) proposes an adaptive and continuous prompting approach to alleviate issues of selection mismatches and limited prompt shareability. MoE-Adapters (Yu et al., 2024) insert a mixture of adapters into the image encoder, activating a few for each task. However, these methods typically involve an auxiliary parameter selection step during inference to extract image features using the expected prompts or adapters, which can lead to misassignments and degrade classification performance. Classifier-based methods, such as RAIL (Xu et al., 2024), extend the classifier dimension while keeping feature representations fixed. It relies on the vanilla zero-shot CLIP to distinguish whether a task has been learned, which can lead to error propagation and degrade the performance.

## 3. Method

### 3.1. Preliminary

**Problem Setting.** Cross-domain Task-Agnostic Incremental Learning (X-TAIL) (Xu et al., 2024) is defined as the scenario where given a pre-trained vision-language model, the learner is required to incrementally learn $K$ different tasks $\{\mathcal{T}^1, \mathcal{T}^2, ..., \mathcal{T}^K\}$. Each task $\mathcal{T}^k = (\mathcal{D}^k, C^k)$ is available only during the $k$-th learning step, where $\mathcal{D}^k = \bigcup_{j=1}^{M^k} \mathcal{D}_j^k$ with $M^k$ denotes the number of classes for task $\mathcal{T}^k$ and $\mathcal{D}_j^k$ denotes the samples of the $j$-th class for task $\mathcal{T}^k$.

$C^k = \{c_j^k\}_{j=1}^{M^k}$ denotes the classnames of the task $\mathcal{T}^k$.

During inference at all steps, the learner attempts to classify input images from any task without the task-identity hint. Formally, the ground-truth label of the test image belongs to $C = C_L \cup C_U$, where $C_L = \bigcup_{j=1}^{k} C^j$ is the union of seen class labels from all previous learning steps and $C_U$ is the set of unseen class labels.

**CLIP Model.** The CLIP (Radford et al., 2021) model contains an image encoder $f_I$ and a text encoder $f_T$. The training of the CLIP model is based on contrastive learning, which aligns image and text features in a shared latent space. This training paradigm enables CLIP to perform zero-shot classification by aligning image and text representations effectively. The zero-shot inference process of the CLIP model for image classification is as follows. First, each class $j \in [M^i]$ of the task $\mathcal{T}^i$ is transformed into a sentence $l_j^i$ using a template "a photo of a $\{c_j^i\}$." Then, the text encoder $f_T$ processes $l_j^i$ to a text feature $t_j^i$, given by $f_T(l_j^i)$. Similarly, the image encoder $f_I$ processes the input image $v$, generating an image embedding $i$. In the following text, superscript numbers are used as task indices, while subscript numbers represent category indices. If a third index is required, it is enclosed in parentheses.

**Text Encoder Fine-tuning Framework for Continual Learning.** Given that CLIP is an efficient continual learner, it can achieve strong classification performance with only a single classifier. Moreover, due to CLIP's strong text-image alignment capabilities, a straightforward yet effective approach is to fine-tune the text encoder and extract text features from class names as the classifier. Specifically, we fine-tune the text encoder to extract optimized text features for the current task and concatenate them with the text features of previous classes to obtain the final classifier. We employ a simple mechanism to prevent catastrophic forgetting. For the current task $\mathcal{T}^k$, the text features $t$ from the previous $k - 1$ tasks are kept frozen, while only the text features for $\mathcal{T}^k$ are updated by gradient optimization.

The objective of continual learning is to minimize the error not only in the current task $\mathcal{T}^k$, but also across all previously learned $\mathcal{T}^{k-1}$ tasks. The optimization objective for fine-tuning the text encoder is formulated as follows:

$$\mathcal{L}(t; k) = \sum_{i=1}^{k} \sum_{j=1}^{M^i} \mathcal{L}(t; k, i, j). \tag{1}$$

The function $\mathcal{L}(t; k, i, j)$ represents the classification loss for the $j$-th class of task $\mathcal{T}^i$ and is defined as follows:

$$\mathbb{E}_{v_j^i \sim \mathbb{P}(\cdot|c_j^i)} \left[ -\log \frac{e^{\langle f_I(v_j^i), t_j^i \rangle}}{\sum_{n \in [k]} \sum_{m \in [M^n]} e^{\langle f_I(v_j^i), t_m^n \rangle}} \right]. \tag{2}$$

For the current task $k$, the estimated loss $\widehat{\mathcal{L}}(\boldsymbol{t}; k, k, j)$ is calculated as follows:

$$\frac{1}{|\mathcal{D}_j^k|} \sum_{\boldsymbol{v} \in \mathcal{D}_j^k} - \log \frac{e^{\langle f_I(\boldsymbol{v}), \boldsymbol{t}_j^k \rangle}}{\sum_{n \in [k]} \sum_{m \in [M^n]} e^{\langle f_I(\boldsymbol{v}), \boldsymbol{t}_m^n \rangle}}. \qquad (3)$$

For classes of previous tasks, the original images are no longer accessible. To address this, we distill $\lambda$ cluster centers, denoted as $\boldsymbol{p}_j^i = \{\boldsymbol{p}_j^i(1), \ldots, \boldsymbol{p}_j^i(\lambda)\}$, which represent distillation image prototypes in $\mathcal{D}_j^i$, to compute the estimated loss $\widehat{\mathcal{L}}(\boldsymbol{t}; k, i, j)$ as follows:

$$\frac{1}{\lambda} \sum_{l=1}^{\lambda} - \log \frac{e^{\langle \boldsymbol{p}_j^i(l), \boldsymbol{t}_j^i \rangle}}{\sum_{n \in [k]} \sum_{m \in [M^n]} e^{\langle \boldsymbol{p}_j^i(l), \boldsymbol{t}_m^n \rangle}}. \qquad (4)$$

The total training loss is the sum of the current task loss in Eq. (3) and the distillation loss in Eq. (4), enabling joint optimization of both current task and past tasks.

Our text encoder fine-tune framework generally preserves the alignment between image and text features, achieving good classification performance, and improving training efficiency by avoiding backpropagation through the image encoder. However, it still struggles with the limited adaptability of image features, which poses a significant challenge in dynamically scaling the training parameters. To address this, we propose a scalable label-specific CLIP adapter to enhance the adaptation capability of image features.

### 3.2. Scalable Label-Specific CLIP Adapter

**Constructing Label-Specific Features.** During the training stage, previous prompt-based methods and MoE-Adapters tend to learn features for specific tasks by selectively activating a partial set of parameters. However, these features are not well aligned with previously learned tasks, necessitating the selective activation of parameters during the inference stage as well, which can lead to suboptimal performance. To address the above issues, we propose LADA, which condenses all task information into a unified representation space during the training stage. This approach eliminates the need for selective parameter activation during inference and helps prevent error propagation. Specifically, LADA aims to generate distinguishing features that capture the specific features of each task to facilitate its discrimination process. To achieve this, LADA employs clustering techniques that have been widely used as standalone tools to gain insights into the properties of data. Given the training set $\mathcal{D}^k = \bigcup_{j=1}^{M^k} \mathcal{D}_j^k$ of the task $\mathcal{T}^k$, we extract all image features of $\mathcal{D}_j^k$ and apply $k$-means clustering to obtain $\lambda_1$ cluster centers $\boldsymbol{W}_j^k \in \mathbb{R}^{\lambda_1 \times d} = \left[\boldsymbol{w}_j^k(1), \ldots, \boldsymbol{w}_j^k(\lambda_1)\right]$. The retained cluster centers characterize the underlying structure of the corresponding class feature space. Therefore, we can construct the label-specific features mapping

$\varphi^k : \boldsymbol{I} \to \mathbb{R}^{M^k \times \lambda_1}$ for task $\mathcal{T}^k$ from the original CLIP image feature space:

$$\varphi^k(\boldsymbol{i}) = \left[\boldsymbol{W}_1^k \boldsymbol{i}, \ldots, \boldsymbol{W}_{M^k}^k \boldsymbol{i}\right]. \qquad (5)$$

Given all above $\varphi^1, \ldots, \varphi^k$, we construct final feature representation by condensing the information across all the tasks $\varphi(\boldsymbol{i}) = [\varphi^1(\boldsymbol{i}), \cdots, \varphi^k(\boldsymbol{i})]$. It is noteworthy that as the number of tasks increases, the features obtained through the aforementioned method can naturally expand. This enhances learning plasticity while ensuring memory stability.

**Training of Label-Specific CLIP Adapter.** Although the initialization process condenses and represents information from all tasks, it does not necessarily guarantee that the extracted features exhibit optimal classification properties. Therefore, we further fine-tune $\boldsymbol{W}$ to enhance the quality of the extracted features. To mitigate catastrophic forgetting, we adopt a simple mechanism by freezing $\boldsymbol{W}^1, \ldots, \boldsymbol{W}^{k-1}$ during fine-tuning on the current task $\mathcal{T}^k$, allowing only $\boldsymbol{W}^k$ to be updated.

To enhance the classification performance of the extracted features, we first consider a fixed classifier $h : \mathbb{R}^{M\lambda_1} \to \mathbb{R}^M$ where $M = \sum_{i=1}^k M^i$ that maps above constructed feature to the corresponding classification logits space, for any $i \in [k]$ and $j \in [M^i]$:

$$(h \circ \varphi)(\boldsymbol{i})_j^i = \phi(\boldsymbol{W}_j^i \boldsymbol{i})\mathbf{1}, \qquad (6)$$

where $\phi = \exp(-\beta(1-x))$ is employed to convert the inner product into non-negative values, with $\beta$ modulating the sharpness of the transformation and $\mathbf{1}$ is a column vector of all ones. Intuitively, the fixed classifier $h$ can be regarded as a nearest-neighbor classifier, which estimates the likelihood that a sample belongs to a certain class by measuring the distance between the sample's features and the representative features of the corresponding class.

For the current task, we can directly minimize the following expected classification cross-entropy loss for each class $j \in [M^k]$ to enhance the classification properties of the extracted features:

$$\frac{1}{|\mathcal{D}_j^k|} \sum_{\boldsymbol{v} \in \mathcal{D}_j^k} - \log \frac{e^{(h \circ \varphi)(f_I(\boldsymbol{v}))_j^k}}{\sum_{n \in [k]} \sum_{m \in [M^n]} e^{(h \circ \varphi)(f_I(\boldsymbol{v}))_m^n}}. \qquad (7)$$

Although the frozen parameters $\boldsymbol{W}^1, \ldots, \boldsymbol{W}^{k-1}$ remain unchanged, ensuring the stability of $(h \circ \varphi)(\boldsymbol{i})_j^i$ for $i \in [k-1]$, which prevents catastrophic forgetting in previous tasks. However, the introduction of the new parameter $\boldsymbol{W}^k$ for the current task may lead to the misclassification of the old task samples into current task classes. Therefore, when fine-tuning $\boldsymbol{W}^k$, it is crucial to maintain a clear distinction between samples from previous tasks and the new task classes. By utilizing the distilled $\lambda_2$ cluster centers, denoted

*Table 1.* Comparison of different continual learning methods on X-TAIL 16-shot for each task in terms of *Transfer*, *Average*, and *Last* scores (%). The best results are highlighted with **bold** style.

| Method | Aircraft | Caltech101 | DTD | EuroSAT | Flowers | Food | MNIST | Pets | Cars | Sun397 | Average |
|---|---|---|---|---|---|---|---|---|---|---|---|
| Zero-shot | 23.8 | 74.3 | 36.4 | 37.4 | 64.1 | 83.4 | 43.9 | 87.8 | 65.5 | 60.8 | 57.7 |
| **Transfer** | | | | | | | | | | | |
| LwF (Li & Hoiem, 2017) | – | 66.6 | 26.9 | 19.5 | 51.0 | 78.4 | 26.6 | 68.9 | 35.5 | 56.1 | 47.7 |
| WiSE-FT (Wortsman et al., 2022b) | – | 70.1 | 31.9 | 25.3 | 56.3 | 79.8 | 29.9 | 74.9 | 45.6 | 56.8 | 52.3 |
| ZSCL (Zheng et al., 2023) | – | 73.3 | 32.6 | **36.8** | 62.1 | 83.8 | **42.1** | 83.6 | 56.5 | 60.2 | 59.0 |
| MoE-Adapters (Yu et al., 2024) | – | 71.0 | 34.9 | 19.2 | 63.0 | **86.6** | 20.0 | 87.2 | 63.7 | 58.6 | 56.0 |
| Ours | – | **75.0** | **36.1** | 35.9 | **66.3** | 83.7 | **42.1** | **88.0** | 65.3 | **61.4** | **61.5** |
| **Average** | | | | | | | | | | | |
| LwF (Li & Hoiem, 2017) | 24.7 | 79.7 | 38.3 | 36.9 | 63.9 | 81.0 | 36.5 | 71.9 | 42.7 | 56.7 | 53.2 |
| WiSE-FT (Wortsman et al., 2022b) | 27.1 | 76.5 | 40.9 | 31.3 | 68.7 | 81.6 | 31.4 | 74.7 | 51.7 | 58.4 | 54.2 |
| ZSCL (Zheng et al., 2023) | 36.0 | 75.0 | 40.7 | 40.5 | 71.0 | 85.3 | 46.3 | 83.3 | 60.7 | 61.5 | 60.0 |
| MoE-Adapters (Yu et al., 2024) | 43.6 | 77.9 | 52.1 | 34.7 | 75.9 | **86.3** | 45.2 | 87.4 | 66.6 | 60.2 | 63.0 |
| Primal-RAIL (Xu et al., 2024) | 44.6 | 89.5 | 56.1 | 69.0 | 83.8 | 85.0 | 63.2 | 88.9 | 68.6 | 62.2 | 71.1 |
| Dual-RAIL (Xu et al., 2024) | 45.4 | 89.4 | 56.4 | 69.6 | 84.0 | 85.0 | **63.5** | 88.8 | 68.8 | 62.3 | 71.3 |
| Ours | **49.1** | **91.0** | **61.3** | **71.6** | **84.4** | 85.0 | 62.8 | **89.7** | **69.2** | **62.9** | **72.7** |
| **Last** | | | | | | | | | | | |
| LwF (Li & Hoiem, 2017) | 20.9 | 83.1 | 47.5 | 38.2 | 75.5 | 84.7 | 50.1 | 78.0 | 75.8 | 74.6 | 62.8 |
| WiSE-FT (Wortsman et al., 2022b) | 21.8 | 76.8 | 42.9 | 20.8 | 77.5 | 84.9 | 30.7 | 76.6 | 75.8 | 72.5 | 58.0 |
| ZSCL (Zheng et al., 2023) | 33.1 | 75.3 | 43.5 | 35.2 | 74.6 | **87.4** | 50.4 | 84.2 | 77.3 | 73.4 | 63.4 |
| MoE-Adapters (Yu et al., 2024) | 43.2 | 78.7 | 57.6 | 32.8 | 79.4 | 86.0 | 86.7 | 87.8 | 78.2 | 74.2 | 70.5 |
| Primal-RAIL (Xu et al., 2024) | 44.2 | **94.6** | 66.8 | 85.9 | 96.3 | 86.8 | 91.6 | 91.5 | 80.6 | 75.4 | 81.4 |
| Dual-RAIL (Xu et al., 2024) | 45.5 | **94.6** | 68.6 | **87.7** | **97.2** | 86.9 | 92.8 | 91.4 | 81.9 | 75.9 | 82.3 |
| Ours | **49.6** | 93.7 | **69.3** | 86.9 | 96.7 | 86.9 | **93.8** | **93.7** | **84.6** | **76.0** | **83.1** |

as $\boldsymbol{p}_j^i = \{\boldsymbol{p}_j^i(1), \ldots, \boldsymbol{p}_j^i(\lambda_2)\}$, we minimize the following expected classification loss for each previous task $i < k$ and class $j \in [M^i]$:

$$\frac{1}{\lambda_2} \sum_{l=1}^{\lambda_2} - \log \frac{e^{(h \circ \varphi)(\boldsymbol{p}_j^i(l))_j^i}}{\sum_{n \in [k]} \sum_{m \in [M^n]} e^{(h \circ \varphi)(\boldsymbol{p}_j^i(l))_m^n}}. \quad (8)$$

**Distribution-Preserved Training.** For previous tasks, distilling only $\lambda_2$ cluster centers $\boldsymbol{p}_j^i$ is insufficient to preserve sufficient information of $\mathcal{D}_j^i$. To better capture the underlying distribution, we fit $\mathcal{D}_j^i$ using Gaussian Mixture Model:

$$\{\pi_j^i(l), \boldsymbol{p}_j^i(l), \boldsymbol{\Sigma}_i^j(l)\}_{l=1}^{\lambda_2} = \text{GMM}(\mathcal{D}_j^i). \quad (9)$$

Given the estimated parameters, we calculate the classification loss for task $i \in [k-1]$ and $j \in [M^i]$ using the augmented features:

$$\sum_{l=1}^{\lambda_2} -\pi_j^i(l) \log \frac{e^{(h \circ \varphi)(\widetilde{\boldsymbol{p}}_j^i(l))_j^i}}{\sum_{n \in [k]} \sum_{m \in [M^n]} e^{(h \circ \varphi)(\widetilde{\boldsymbol{p}}_j^i(l))_m^n}}, \quad (10)$$

where $\pi$ represents the mixture weight of each component and $\widetilde{\boldsymbol{p}}$ is augmented prototypes defined as follows:

$$\widetilde{\boldsymbol{p}}_j^i(l) = \boldsymbol{p}_j^i(l) + \boldsymbol{e} \cdot \sqrt{\frac{\text{Tr}\left(\boldsymbol{\Sigma}_j^i(l)\right)}{d}}, \quad (11)$$

where $\boldsymbol{e}$ is Gaussian noise with the same dimension as the prototype and the scale $\sqrt{\frac{\text{Tr}(\boldsymbol{\Sigma}_j^i(l))}{d}}$ controls the uncertainty of the augmented prototypes.

**Overall Framework.** During the training stage, we jointly optimize the text encoder and the LADA feature extraction module. Specifically, the cross-entropy loss is computed by summing the logits produced by the text encoder, which are derived from Eq. (3) and Eq. (4), and the logits produced by LADA, which are derived from Eq. (7) and Eq. (10). By integrating these two sets of logits, the model is encouraged to leverage both textual information and visual feature adaptations comprehensively. This training approach facilitates an effective end-to-end learning process through their simultaneous optimization.

During the inference stage, for all seen classes $C_L$, we utilize their text features as classification features, while for unseen classes $C_U$, the text features are extracted using the vanilla text encoder and also serve as classification features. These classification features are concatenated for the final classification. If the predicted class belongs to the unseen classes $C_U$, the text-visual classification result is used directly. Otherwise, for classes in $C_L$, the final prediction is obtained by applying a linear weighting between the logits produced by LADA and the corresponding text features.

*Table 2.* Comparison of different continual learning methods on X-TAIL full-shot for each task in terms of *Transfer*, *Average*, and *Last* scores (%). The best results are highlighted with **bold** style.

| Method | Aircraft | Caltech101 | DTD | EuroSAT | Flowers | Food | MNIST | Pets | Cars | Sun397 | Average |
|---|---|---|---|---|---|---|---|---|---|---|---|
| Zero-shot | 23.8 | 74.3 | 36.4 | 37.4 | 64.1 | 83.4 | 43.9 | 87.8 | 65.5 | 60.8 | 57.7 |
| **Transfer** | | | | | | | | | | | |
| LwF (Li & Hoiem, 2017) | – | 62.4 | 27.8 | 10.7 | 52.0 | 76.0 | 25.4 | 68.3 | 30.4 | 54.5 | 45.3 |
| WiSE-FT (Wortsman et al., 2022b) | – | 59.9 | 25.4 | 10.2 | 43.9 | 67.9 | 29.4 | 57.4 | 24.1 | 50.3 | 40.9 |
| ZSCL (Zheng et al., 2023) | – | 71.3 | 33.8 | 33.0 | **66.2** | 85.2 | 40.2 | 81.9 | 57.3 | **62.5** | 59.0 |
| MoE-Adapters (Yu et al., 2024) | – | 69.5 | 30.8 | 19.0 | 60.3 | **85.6** | 43.7 | 85.6 | 55.5 | 57.3 | 56.4 |
| Ours | – | **75.2** | **36.1** | **36.7** | 65.6 | 83.9 | **45.2** | **88.0** | 65.3 | 61.1 | **61.9** |
| **Average** | | | | | | | | | | | |
| LwF (Li & Hoiem, 2017) | 25.8 | 81.2 | 48.1 | 33.1 | 57.0 | 75.1 | 54.9 | 74.5 | 35.5 | 57.1 | 54.2 |
| WiSE-FT (Wortsman et al., 2022b) | 14.9 | 79.8 | 45.3 | 17.1 | 55.7 | 70.1 | 52.0 | 68.0 | 35.6 | 53.3 | 49.2 |
| ZSCL (Zheng et al., 2023) | 39.7 | 80.8 | 52.9 | 40.8 | 79.3 | **88.0** | 51.4 | 85.5 | 62.9 | **64.1** | 64.5 |
| MoE-Adapters (Yu et al., 2024) | 52.4 | 79.4 | 57.7 | 42.7 | 81.1 | 86.6 | 64.8 | 86.7 | 61.3 | 59.0 | 67.2 |
| Primal-RAIL (Xu et al., 2024) | 48.8 | 89.6 | 59.0 | 74.4 | 84.0 | 86.1 | **65.6** | 89.5 | 68.9 | 62.3 | 72.8 |
| Ours | **53.9** | **93.6** | **66.6** | **78.0** | **85.3** | 86.7 | 65.2 | **89.9** | **69.7** | 62.7 | **75.2** |
| **Last** | | | | | | | | | | | |
| LwF (Li & Hoiem, 2017) | 9.6 | 77.1 | 55.3 | 38.7 | 60.5 | 83.1 | **99.5** | 85.9 | 49.6 | 80.0 | 63.9 |
| WiSE-FT (Wortsman et al., 2022b) | 18.1 | 84.9 | 53.4 | 27.0 | 69.6 | 88.0 | 88.4 | 91.5 | 76.7 | **80.2** | 67.8 |
| ZSCL (Zheng et al., 2023) | 33.8 | 80.4 | 60.2 | 31.1 | 85.8 | **91.3** | 80.4 | 93.7 | 84.9 | 79.0 | 72.1 |
| MoE-Adapters (Yu et al., 2024) | 51.9 | 79.0 | 64.2 | 51.5 | 95.1 | 87.6 | 96.4 | 89.1 | 84.4 | 74.0 | 77.3 |
| Primal-RAIL (Xu et al., 2024) | 45.8 | 94.1 | 70.7 | 94.2 | 96.5 | 89.0 | 98.1 | 93.5 | 82.0 | 76.5 | 84.0 |
| Ours | **55.5** | **96.2** | **75.8** | **95.8** | **98.4** | 89.6 | 98.8 | **94.5** | **87.3** | 77.2 | **86.9** |

## 4. Experiments

**Benchmarks.** We conduct experiments on the recently proposed X-TAIL (Xu et al., 2024) benchmark which consists of 10 image classification datasets: Aircraft (Maji et al., 2013), Caltech101 (Fei-Fei et al., 2004), DTD (Cimpoi et al., 2014), EuroSAT (Helber et al., 2019), Flowers (Nilsback & Zisserman, 2008), Food (Bossard et al., 2014), MNIST (Deng, 2012), OxfordPet (Parkhi et al., 2012), StanfordCars (Krause et al., 2013), and SUN397 (Xiao et al., 2010). Each dataset is treated as a task, and the benchmark includes a total of 1,100 classes across the 10 tasks.

In addition to the 16-shot setting proposed by (Xu et al., 2024), in which 16 training samples per class were selected for each task, we also evaluate the benchmark under a full-shot setting. This more realistic scenario maintains the original dataset distribution, with varying numbers of training samples across tasks, providing a more comprehensive and challenging evaluation for continual learning methods.

**Evaluation Metrics.** To evaluate both *stability* and *plasticity* as discussed in Section 1, we employ the *Transfer*, *Average*, and *Last* metrics from Zheng et al. (2023). The *Last* metric assesses model performance after continual training, capturing plasticity and backward forgetting. To quantify forward forgetting, we calculate the model's average accuracy on tasks $k + 1, k + 2, \ldots, K$ after training on task $k$,

defining the *Transfer* metric. The *Average* metric represents the mean accuracy across all time steps, offering a holistic measure of stability and plasticity. Detailed definitions of these metrics are provided in Appendix A.

**Implementation Details.** We adopt the CLIP (Radford et al., 2021) model with a ViT-B/16 (Dosovitskiy et al., 2021) image encoder. The training process is carried out using the AdamW (Loshchilov & Hutter, 2019) optimizer, with a learning rate of 0.001 and a batch size of 64 across all tasks. For the primary experiments, we set the hyperparameters as $\lambda_1 = 16$ and $\lambda_2 = 4$. All experiments of LADA are conducted on a single NVIDIA 4090 GPU. Additional implementation details for the fine-tuning of the text encoder are provided in Appendix B.

### 4.1. Main Results

We evaluate our method on both 16-shot and full-shot settings. The learning order is set alphabetically: Aircraft, Caltech101, DTD, EuroSAT, Flowers, Food, MNIST, OxfordPet, StanfordCars, and SUN397. Additional experiments in random order are provided in Appendix C. The performance averaged across the 10 tasks for our method and other baseline approaches in the X-TAIL setting are presented in the *Average* column of both Table 1 and Table 2. The *Zero-shot* indicates the zero-shot performance of the pre-trained CLIP model on each task.

*Table 3.* Ablation study of LADA in 16-shot and full-shot settings.

| Settings | BF | DPT | LADA | Transfer | Average | Last |
|---|---|---|---|---|---|---|
| 16-shot | √ | | | 59.4 | 70.9 | 82.1 |
| | √ | √ | | 59.9 | 71.6 | 82.6 |
| | √ | | √ | 61.2 | 72.3 | 83.0 |
| | √ | √ | √ | **61.5** | **72.7** | **83.1** |
| Full-shot | √ | | | 59.6 | 73.2 | 85.6 |
| | √ | √ | | 60.3 | 74.4 | 86.3 |
| | √ | | √ | 61.1 | 74.9 | 86.8 |
| | √ | √ | √ | **61.9** | **75.2** | **86.9** |

In the 16-shot setting, our method outperforms the previous best approach with a 2.5% increase in *Transfer* accuracy, a 1.4% increase in *Average* accuracy, and a 0.8% increase in *Last* accuracy. In the full-shot setting, which represents a more realistic and challenging scenario, our method achieves even greater gains over the baseline, with improvements of 2.9% in *Transfer* accuracy, 2.4% in *Average* accuracy, and 2.9% in *Last* accuracy. These results indicate that our approach enhances both stability and plasticity, effectively learning knowledge from new tasks and preserving them while retaining pre-trained knowledge. This is further supported by Figure 2, which provides a clear visualization of the accuracy changes across all tasks over all learning steps. Notably, on datasets such as Flowers, Food, and SUN397, our method surpasses the vanilla zero-shot CLIP model in *Transfer* accuracy, indicating that it not only mitigates forward forgetting, but also enhances the retention of pre-trained knowledge. We further analyze the underlying reasons for this improvement in Section 4.5.

We do not present the performance of RAIL on the *Transfer* metric, as their reported results directly use the zero-shot accuracy of the vanilla CLIP model, which does not reflect the extent of forward forgetting on model after continual learning. Additionally, since Dual-RAIL needs to store all features, it is not possible to complete training under the full-shot setting because of computational overhead and GPU memory constraint.

## 4.2. Ablation Study

We analyze the contributions of our basic text encoder fine-tuning framework (BF), distribution-preserved training (DPT), and LADA under both 16-shot and full-shot settings, with consistent findings presented in Table 3. The BF setup, which fine-tunes only the text encoder, achieves competitive baseline performance. However, it does not effectively address plasticity and stability due to the absence of adaptation of image features and learned distribution preservation. Incorporating LADA improves both the *Average* and *Last* metrics by facilitating the learning of better label-specific feature representations. Additionally, LADA enhances the *Transfer* metric by mitigating forward forgetting, reducing

*Table 4.* Impact of label-specific dimension $\lambda_1$ and prototype number $\lambda_2$ on TAIL full-shot setting. The asterisk (*) indicates the number of samples in some classes is fewer than 32. *Time* is measured in seconds per batch (s/batch), *Memory* is measured in gigabytes (GB), and *Params* is measured in millions (M).

| $\lambda_1$ | $\lambda_2$ | Transfer | Average | Last | Time | Memory | Params |
|---|---|---|---|---|---|---|---|
| 8 | 1 | 60.9 | 74.7 | 86.7 | 0.280 | 17.84 | 4.51 |
| 8 | 4 | 61.7 | 75.1 | 86.7 | 0.287 | 18.14 | 4.51 |
| 8 | 16 | 62.2 | 75.1 | 86.7 | 0.318 | 19.05 | 4.51 |
| 16 | 1 | 61.4 | 75.1 | 86.9 | 0.281 | 18.01 | 9.01 |
| 16 | 4 | 61.9 | 75.2 | 86.9 | 0.289 | 18.51 | 9.01 |
| 16 | 16 | 62.3 | 74.8 | 86.9 | 0.330 | 20.62 | 9.01 |
| 32* | 1 | 60.9 | 74.7 | 86.9 | 0.282 | 18.42 | 17.51 |
| 32* | 4 | 61.2 | 74.6 | 86.9 | 0.297 | 18.97 | 17.51 |
| 32* | 16 | 61.9 | 74.0 | 86.9 | 0.358 | 23.13 | 17.51 |

semantic drift introduced during text encoder fine-tuning. The DPT module further strengthens the representations of image prototypes by generating underlying distributions, leading to overall performance gains. This ablation study highlights the effectiveness of LADA in improving both stability and plasticity.

## 4.3. Analysis of LADA Dimension and Prototypes

We analyze the impact of different LADA dimensions and the number of distilled prototypes per class on both performance and computational cost in the full-shot setting, with results presented in Table 4.

**Performance Impact.** Adjusting the LADA dimension has minimal effect on performance metrics, demonstrating the model's robustness to this parameter. Distilling a single feature as prototype per class, combined with DPT, is sufficient to prevent backward forgetting, as reflected in the stable *Last* metric. However, increasing the number of prototypes improves distribution estimation for each class in previous tasks, reducing the risk of semantic feature space shifts caused by image prototype augmentation, which helps preserve pre-trained knowledge and prevents degradation in *Transfer* performance.

**Computational Cost.** We evaluate computational costs by measuring the time cost per batch for the final task (which involves the most sampled image prototypes), peak memory usage during continual learning across all $K$ tasks, and the total parameter count of LADA. Since previously trained portions of LADA remain frozen, only $1/K$ of its total parameters are updated on average per task. Our results show that, under the same prototype settings, increasing the LADA dimension has minimal impact on both time and memory costs. This is because LADA adjusts features after the backbone outputs, eliminating the need for back-propagation through the image encoder. When the LADA dimension is fixed, increasing the number of prototypes per

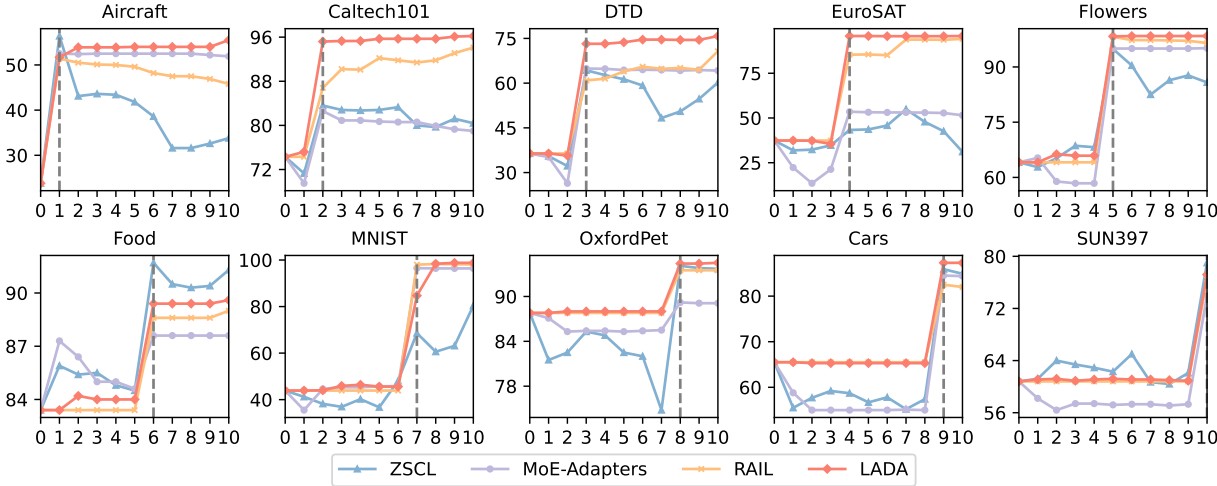

*Figure 2.* Accuracy (%) changes across all tasks over all learning steps in the full-shot setting.

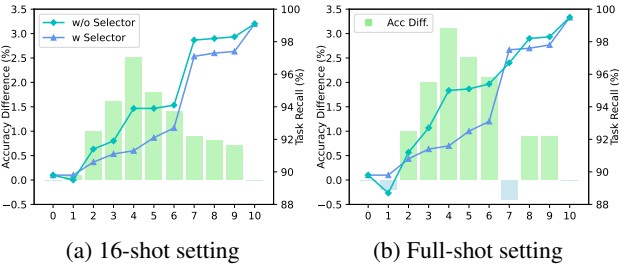

(a) 16-shot setting  (b) Full-shot setting

*Figure 3.* Comparison of whether to use zero-shot CLIP as a selector to distinguish between seen and unseen classes. Without the selector, LADA performs better in the continual learning process as it utilizes the learned knowledge to improve average task recall.

class leads to moderate growth in time and memory costs. However, distilling a small number of prototypes is sufficient for strong performance, making this a cost-effective strategy for mitigating forgetting.

Moreover, the slight increase in time and memory costs with respect to parameters of LADA indicates that it can scale efficiently to accommodate more tasks in continual learning, with only a gradual and accessible increase in resource consumption.

### 4.4. Impacts on Forward Forgetting

In this subsection, we analyze the impact of LADA and DPT on improving *Transfer* performance.

**Impact of LADA.** As shown in Table 3, LADA achieves a significant improvement in *Transfer* without the DPT module. By effectively capturing label-specific features, LADA mitigates semantic drift that typically arises from fine-tuning the text encoder. This reduction in semantic drift helps preserve previously learned knowledge, thereby minimizing forward forgetting and enhancing stability.

**Impact of DPT.** As shown in Table 3, incorporating DPT leads to a moderate improvement in *Transfer*. This is because DPT enhances the model's ability to better preserve the original feature distribution. Furthermore, as shown in Table 4, increasing the number of image prototypes further improves *Transfer* performance. A higher number of prototypes enables a more accurate estimation of embeddings from previous tasks, reducing interference with pre-trained knowledge and mitigating feature space degradation.

### 4.5. Why *Transfer* in Some Tasks Surpasses Zero-Shot CLIP Performance

In this section, we analyze why the *Transfer* metric in some tasks surpasses zero-shot CLIP performance. Figure 3 presents a comparative analysis of whether to use vanilla zero-shot CLIP as a selector to distinguish between seen and unseen classes. The evaluation considers two key performance metrics per training step: task recall which measures the proportion of relevant samples successfully assigned to their respective tasks, and accuracy difference which measures the difference in classification accuracy between LADA and LADA with the selector. LADA achieves higher accuracy in continual learning without relying on vanilla zero-shot CLIP as a selector. In contrast, the RAIL method (Xu et al., 2024) employs CLIP as a selector, restricting the trained model to classifying only the seen classes.

Our unified and straightforward method LADA eliminates the two-step process required by RAIL, reducing error propagation and fully leveraging learned knowledge to distinguish between seen and unseen classes. Notably, our results provide insight into why some task transfers outperform zero-shot CLIP: Since task recall constrains the upper bound on classification accuracy, our method enhances differentiation between seen and unseen classes based on learned

knowledge, thus improving task recall for unseen classes and ultimately improving their classification accuracy. Our experiments show that on datasets such as Flowers, Food, and Sun397, our method outperforms vanilla zero-shot CLIP on *Transfer* metric, as evidenced by Table 1 and Table 2. In real-world continual learning scenarios, it is vital to leverage learned knowledge to enhance generalization performance. Our method effectively achieves this by providing reliable and efficient classification without the complexity of separate classification stages.

## 5. Conclusion

This paper introduces LADA, a novel CLIP adapter that transforms CLIP features into label-specific features by learning a compact set of vectors for each class. LADA maintains high discriminability between seen and new classes by training on both current task samples and distilled representations from previous tasks. Unlike previous CLIP-based approaches, our method condenses all task information into a unified representation space, eliminating the need for auxiliary prompt selection, adapter selection, or reliance on vanilla zero-shot CLIP as an unseen class selector. Extensive empirical results show that our method significantly enhances stability and plasticity in continual learning settings. Moreover, its efficiency enables seamless scaling to more tasks and larger training sets while maintaining a moderate computational cost.

## Acknowledgements

This work was supported by the National Science Foundation of China (62206049, 62225602), and the Big Data Computing Center of Southeast University. We would like to thank anonymous reviewers for their constructive suggestions.

## Impact Statement

This paper presents work whose goal is to advance the field of Machine Learning. There are many potential societal consequences of our work, none which we feel must be specifically highlighted here.

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

## A. Evaluation Metrics

We define the *Transfer*, *Average*, and *Last* metrics to evaluate model performance under continual learning scenarios. Let $\hat{a}_k^{(j)}$ represent the accuracy of the model on task $k$ after training on task $j$. The *Transfer*, *Average*, and *Last* metrics for task $k$ are computed as follows:

$$\text{Transfer}_k = \frac{1}{k-1} \sum_{j=1}^{k-1} \hat{a}_k^{(j)}, \quad k = 2, 3, \ldots, K, \tag{12}$$

$$\text{Average}_k = \frac{1}{K} \sum_{j=1}^{K} \hat{a}_k^{(j)}, \quad k = 1, 2, \ldots, K, \tag{13}$$

$$\text{Last}_k = \hat{a}_k^{(K)}, \quad k = 1, 2, \ldots, K. \tag{14}$$

Here, $K$ denotes the total number of tasks. The *Transfer* metric evaluates the model's ability to retain zero-shot capability by calculating its average accuracy on future tasks $(j + 1, j + 2, \ldots, K)$ after training on task $j$. The *Last* metric measures the model's final performance on each task after completing all training, thereby quantifying the extent of backward forgetting. Finally, the *Average* metric represents the mean accuracy across all time steps, providing a comprehensive view of the model's performance throughout the entire continual learning process.

## B. Additional Implementation Details

To mitigate disruptions to the text encoder and enhance training efficiency, we adopt the parameter-efficient fine-tuning method, AdaptFormer (Chen et al., 2022). This approach allows for the fine-tuning of only a small subset of parameters, while keeping the text encoder backbone frozen. The AdaptFormer operates as follows:

$$\boldsymbol{X}' = \left( \left( \text{ReLU} \left( \text{LN} \left( \boldsymbol{X} \right) \boldsymbol{W}_{\text{down}} \right) \right) \boldsymbol{W}_{\text{up}} + \boldsymbol{X} \right) \cdot s, \tag{15}$$

where $\boldsymbol{X}$ represents the output of the multi-head attention in transformer, $\boldsymbol{W}_{\text{down}}$ and $\boldsymbol{W}_{\text{up}}$ are learnable projection matrices, and $s$ is a learnable scaling factor that controls the impact of each layer on the backbone. AdaptFormer adopts parallel fine-tuning with residual connections from the transformer feedforward network output, enabling minimal interference with the original feature spaces while effectively adapting to the current task.

## C. Comparison of different methods on X-TAIL with order II.

In this section, we compare different methods within the X-TAIL 16-shot setting and full-shot setting using a random order: StanfordCars, Aircraft, OxfordPet, Food, SUN397, MNIST, Flowers, DTD, Caltech101, and EuroSAT. As shown in Table 5 and Table 6, our method consistently outperforms previous approaches across all metrics, further supporting the conclusions drawn in Section 4.1.

## D. Additional X-TAIL Results

In this section, we present the per-training-step accuracies for both the 16-shot and full-shot settings, under Order-I and Order-II, in Tables 7 to 10. These results demonstrate strong performance in terms of both learning plasticity and memory stability.

*Table 5.* Comparison of different continual learning methods on X-TAIL 16-shot for each task with order-II in terms of *Transfer*, *Average*, and *Last* scores (%). The best results are highlighted with **bold** style.

| Method | Cars | Aircraft | Pets | Food | SUN397 | MNIST | Flowers | DTD | Caltech101 | EuroSAT | Average |
|---|---|---|---|---|---|---|---|---|---|---|---|
| Zero-shot | 65.5 | 23.8 | 87.8 | 83.4 | 60.8 | 43.9 | 64.1 | 36.4 | 74.3 | 37.4 | 57.7 |
| **Transfer** | | | | | | | | | | | |
| LwF (Li & Hoiem, 2017) | – | 20.0 | 74.1 | 79.6 | 58.1 | 34.1 | 48.9 | 27.7 | 64.4 | 15.1 | 46.9 |
| WiSE-FT (Wortsman et al., 2022b) | – | 21.3 | 79.5 | 83.3 | 61.0 | 39.9 | 56.5 | 29.6 | 68.0 | 20.8 | 51.1 |
| ZSCL (Zheng et al., 2023) | – | 23.0 | 84.3 | 87.2 | **63.0** | 42.1 | 65.2 | 34.6 | 71.4 | **40.9** | **56.9** |
| MoE-Adapters (Yu et al., 2024) | – | 17.1 | 87.2 | **87.5** | 58.4 | 12.6 | 65.5 | 35.9 | 70.0 | 17.9 | 50.2 |
| Ours | – | **23.8** | **87.8** | 84.3 | 61.3 | **42.2** | **65.7** | **37.2** | **71.9** | 36.0 | 56.7 |
| **Average** | | | | | | | | | | | |
| LwF (Li & Hoiem, 2017) | 49.0 | 27.4 | 69.7 | 83.0 | 65.7 | 42.2 | 63.5 | 33.1 | 68.5 | 17.5 | 52.0 |
| WiSE-FT (Wortsman et al., 2022b) | 57.9 | 29.6 | 77.8 | 85.4 | 68.0 | 51.6 | 69.3 | 35.5 | 71.0 | 23.0 | 56.9 |
| ZSCL (Zheng et al., 2023) | 74.4 | 36.4 | 86.7 | **88.7** | 68.9 | 50.0 | 75.1 | 40.1 | 72.5 | **43.7** | 63.6 |
| MoE-Adapters (Yu et al., 2024) | 74.4 | 38.6 | 87.7 | 87.3 | 67.9 | 50.6 | 76.5 | 43.7 | 72.3 | 18.8 | 61.8 |
| Primal-RAIL (Xu et al., 2024) | 81.1 | 43.0 | 91.1 | 86.6 | 69.0 | 59.2 | 76.7 | 45.6 | **78.4** | 42.3 | 67.3 |
| Dual-RAIL (Xu et al., 2024) | 82.3 | 43.9 | 91.0 | 85.6 | 69.3 | 59.5 | 77.1 | 45.8 | **78.4** | 42.5 | 67.5 |
| Ours | **84.7** | **46.2** | **92.2** | 86.4 | **70.0** | **68.0** | 77.4 | **47.2** | 75.9 | 41.2 | **68.9** |
| **Last** | | | | | | | | | | | |
| LwF (Li & Hoiem, 2017) | 29.6 | 17.5 | 63.0 | 83.8 | 67.7 | 44.9 | 79.3 | 44.8 | 84.6 | 39.0 | 55.4 |
| WiSE-FT (Wortsman et al., 2022b) | 46.1 | 23.5 | 71.3 | 85.7 | 70.2 | 59.1 | 85.5 | 47.9 | 82.4 | 42.8 | 61.5 |
| ZSCL (Zheng et al., 2023) | 71.7 | 35.3 | 86.5 | **89.2** | 71.8 | 52.3 | 89.8 | 52.0 | 77.1 | 68.4 | 69.4 |
| MoE-Adapters (Yu et al., 2024) | 75.1 | 41.1 | 87.9 | 87.1 | 74.1 | 89.7 | 92.6 | 61.2 | 81.0 | 27.4 | 71.7 |
| Primal-RAIL (Xu et al., 2024) | 80.8 | 45.0 | 92.3 | 86.7 | 75.1 | 91.8 | 96.4 | 68.2 | 94.6 | 86.9 | 81.8 |
| Dual-RAIL (Xu et al., 2024) | 82.3 | 46.3 | 92.2 | 86.9 | 75.7 | 92.6 | **97.4** | 69.0 | **94.7** | **88.3** | 82.5 |
| Ours | **84.8** | **49.2** | **93.6** | 87.6 | **76.6** | **93.9** | **97.4** | **70.7** | 91.7 | 87.4 | **83.3** |

*Table 6.* Comparison of different continual learning methods on X-TAIL full-shot for each task with order-II in terms of *Transfer*, *Average*, and *Last* scores (%). The best results are highlighted with **bold** style.

| Method | Cars | Aircraft | Pets | Food | SUN397 | MNIST | Flowers | DTD | Caltech101 | EuroSAT | Average |
|---|---|---|---|---|---|---|---|---|---|---|---|
| Zero-shot | 65.5 | 23.8 | 87.8 | 83.4 | 60.8 | 43.9 | 64.1 | 36.4 | 74.3 | 37.4 | 57.7 |
| **Transfer** | | | | | | | | | | | |
| LwF (Li & Hoiem, 2017) | – | 16.7 | 78.5 | 76.0 | 59.7 | 41.3 | 46.6 | 27.3 | 63.3 | 10.4 | 46.6 |
| WiSE-FT (Wortsman et al., 2022b) | – | 20.3 | 77.9 | 75.7 | 55.6 | 39.6 | 45.0 | 25.4 | 58.9 | 8.3 | 45.2 |
| ZSCL (Zheng et al., 2023) | – | 21.7 | 83.2 | 85.6 | **63.0** | 39.3 | 61.8 | 34.3 | **72.2** | 26.4 | 54.2 |
| MoE-Adapters (Yu et al., 2024) | – | 17.5 | 87.1 | **86.8** | 58.2 | **44.2** | 63.4 | 33.9 | 67.9 | 15.3 | 52.7 |
| Ours | – | **23.8** | **87.8** | 84.3 | 61.4 | 41.6 | **65.7** | 34.5 | 71.0 | 28.9 | **55.4** |
| **Average** | | | | | | | | | | | |
| LwF (Li & Hoiem, 2017) | 38.6 | 21.6 | 73.2 | 75.9 | 67.7 | 69.4 | 62.2 | 36.8 | 68.0 | 15.1 | 52.9 |
| WiSE-FT (Wortsman et al., 2022b) | 47.1 | 30.9 | 77.9 | 76.6 | 65.0 | 59.0 | 58.7 | 36.1 | 65.4 | 9.8 | 52.7 |
| ZSCL (Zheng et al., 2023) | 77.8 | 43.1 | 90.3 | **89.5** | **71.8** | 61.3 | 73.6 | 42.9 | 74.2 | 26.8 | 65.1 |
| MoE-Adapters (Yu et al., 2024) | 84.2 | 47.4 | 89.0 | 88.0 | 65.2 | **70.7** | 76.3 | 43.4 | 71.5 | 14.8 | 65.1 |
| Primal-RAIL (Xu et al., 2024) | 83.0 | 45.7 | 92.1 | 87.1 | 70.0 | 61.6 | 76.8 | 46.4 | **78.3** | 43.1 | 68.4 |
| Ours | **87.0** | **49.3** | **93.1** | 88.0 | 70.9 | 66.8 | **79.0** | 46.8 | 76.0 | 35.6 | **69.2** |
| **Last** | | | | | | | | | | | |
| LwF (Li & Hoiem, 2017) | 13.4 | 6.9 | 58.2 | 74.5 | 71.2 | **99.4** | 76.3 | 53.2 | 85.4 | 57.0 | 59.6 |
| WiSE-FT (Wortsman et al., 2022b) | 19.3 | 2.0 | 70.5 | 77.6 | 67.9 | 72.1 | 66.2 | 52.7 | 90.6 | 22.7 | 54.2 |
| ZSCL (Zheng et al., 2023) | 75.6 | 31.7 | 90.4 | **90.7** | 77.0 | 75.4 | 88.3 | 60.5 | 82.1 | 29.6 | 70.1 |
| MoE-Adapters (Yu et al., 2024) | 84.1 | 50.6 | 88.9 | 88.1 | 68.7 | 97.2 | 95.5 | 65.6 | 86.2 | 10.3 | 73.5 |
| Primal-RAIL (Xu et al., 2024) | 81.9 | 46.1 | 93.3 | 89.0 | 76.6 | 98.2 | 96.6 | 70.6 | 94.1 | 94.1 | 84.1 |
| Ours | **87.0** | **55.4** | **94.5** | 89.6 | **77.7** | 98.8 | **99.0** | 75.8 | **95.8** | **95.8** | **86.9** |

*Table 7.* Accuracy of Ours on the X-TAIL 16-shot with order-I. Each row represents the performance on every dataset of the model trained after the corresponding task. `Transfer` , `Average` , and `Last` metrics are shown.

| | Aircraft | Caltech101 | DTD | EuroSAT | Flowers | Food | MNIST | Pets | Cars | SUN397 | |
|---|---|---|---|---|---|---|---|---|---|---|---|
| Transfer | | 75.0 | 36.1 | 35.9 | 66.3 | 83.7 | 42.1 | 88.0 | 65.3 | 61.4 | 61.5 |
| Aircraft | 48.6 | 75.0 | 36.4 | 37.4 | 64.1 | 83.4 | 43.9 | 87.8 | 65.5 | 61.1 | |
| Caltech101 | 49.1 | 91.7 | 35.7 | 37.1 | 67.2 | 83.9 | 44.0 | 88.0 | 65.3 | 61.4 | |
| DTD | 49.1 | 92.5 | 66.7 | 33.1 | 67.0 | 83.7 | 44.5 | 88.0 | 65.3 | 61.2 | |
| EuroSAT | 49.1 | 92.5 | 66.7 | 86.9 | 67.0 | 83.7 | 40.1 | 88.0 | 65.3 | 61.4 | |
| Flowers | 49.1 | 92.7 | 66.8 | 86.9 | 96.3 | 83.7 | 40.1 | 88.0 | 65.3 | 61.5 | |
| Food | 49.1 | 92.7 | 67.8 | 86.9 | 96.4 | 86.1 | 40.1 | 88.0 | 65.3 | 61.5 | |
| MNIST | 49.1 | 92.9 | 67.8 | 86.9 | 96.4 | 86.2 | 93.8 | 88.0 | 65.3 | 61.5 | |
| Pets | 49.1 | 93.0 | 67.9 | 86.9 | 96.4 | 86.2 | 93.8 | 93.5 | 65.3 | 61.5 | |
| Cars | 49.1 | 93.2 | 67.9 | 86.9 | 96.4 | 86.2 | 93.8 | 93.5 | 84.6 | 61.6 | |
| SUN397 | 49.6 | 93.7 | 69.3 | 86.9 | 96.7 | 86.9 | 93.8 | 93.7 | 84.6 | 76.0 | 83.1 |
| Average | 49.1 | 91.0 | 61.3 | 71.6 | 84.4 | 85.0 | 62.8 | 89.7 | 69.2 | 62.9 | 72.7 |

*Table 8.* Accuracy of Ours on the X-TAIL full-shot with order-I. Each row represents the performance on every dataset of the model trained after the corresponding task. `Transfer` , `Average` , and `Last` metrics are shown.

| | Aircraft | Caltech101 | DTD | EuroSAT | Flowers | Food | MNIST | Pets | Cars | SUN397 | |
|---|---|---|---|---|---|---|---|---|---|---|---|
| Transfer | | 75.2 | 36.1 | 36.7 | 65.6 | 83.9 | 45.2 | 88.0 | 65.3 | 61.1 | 61.9 |
| Aircraft | 51.7 | 75.2 | 36.4 | 37.4 | 64.1 | 83.4 | 43.9 | 87.8 | 65.5 | 61.1 | |
| Caltech101 | 53.9 | 95.2 | 35.7 | 37.3 | 66.3 | 84.2 | 44.0 | 88.0 | 65.3 | 61.2 | |
| DTD | 53.9 | 95.3 | 73.2 | 35.5 | 65.9 | 84.0 | 45.9 | 88.0 | 65.3 | 60.9 | |
| EuroSAT | 53.9 | 95.3 | 73.2 | 95.8 | 65.9 | 84.0 | 46.4 | 88.0 | 65.3 | 61.1 | |
| Flowers | 54.0 | 95.7 | 73.7 | 95.8 | 98.3 | 84.0 | 45.6 | 88.0 | 65.3 | 61.2 | |
| Food | 54.0 | 95.7 | 74.6 | 95.7 | 98.4 | 89.4 | 45.6 | 88.0 | 65.3 | 61.1 | |
| MNIST | 54.0 | 95.7 | 74.6 | 95.7 | 98.4 | 89.4 | 84.7 | 88.0 | 65.3 | 61.1 | |
| Pets | 54.0 | 95.7 | 74.5 | 95.7 | 98.4 | 89.4 | 98.4 | 94.4 | 65.3 | 61.0 | |
| Cars | 54.0 | 96.1 | 74.5 | 95.7 | 98.4 | 89.4 | 98.8 | 94.4 | 87.3 | 60.9 | |
| SUN397 | 55.5 | 96.2 | 75.8 | 95.8 | 98.4 | 89.6 | 98.8 | 94.5 | 87.3 | 77.2 | 86.9 |
| Average | 53.9 | 93.6 | 66.6 | 78.0 | 85.3 | 86.7 | 65.2 | 89.9 | 69.7 | 62.7 | 75.2 |

*Table 9.* Accuracy of Ours on the X-TAIL 16-shot with order-II. Each row represents the performance on every dataset of the model trained after the corresponding task. Transfer , Average , and Last metrics are shown.

| | Cars | Aircraft | Pets | Food | SUN397 | MNIST | Flowers | DTD | Caltech101 | EuroSAT | |
|---|---|---|---|---|---|---|---|---|---|---|---|
| Transfer | | 23.8 | 87.8 | 84.3 | 61.3 | 42.2 | 65.7 | 37.2 | 71.9 | 36.0 | 56.7 |
| Cars | 84.7 | 23.8 | 87.8 | 84.3 | 61.1 | 43.9 | 65.7 | 36.4 | 72.5 | 37.4 | |
| Aircraft | 84.7 | 48.4 | 87.8 | 84.3 | 61.4 | 43.9 | 65.7 | 36.4 | 73.1 | 37.4 | |
| Pets | 84.7 | 48.4 | 93.2 | 84.3 | 61.4 | 41.0 | 65.7 | 36.4 | 72.8 | 37.4 | |
| Food | 84.7 | 48.4 | 93.2 | 86.8 | 61.4 | 41.0 | 65.7 | 37.9 | 73.3 | 37.4 | |
| SUN397 | 84.7 | 48.6 | 93.3 | 87.1 | 75.3 | 41.0 | 65.7 | 37.6 | 70.3 | 35.3 | |
| MNIST | 84.7 | 48.6 | 93.3 | 87.1 | 75.3 | 93.8 | 65.7 | 37.6 | 70.3 | 35.5 | |
| Flowers | 84.7 | 48.7 | 93.3 | 87.1 | 75.4 | 93.8 | 92.0 | 38.1 | 71.4 | 35.5 | |
| DTD | 84.7 | 48.7 | 93.3 | 87.4 | 75.5 | 93.8 | 92.7 | 70.4 | 71.8 | 34.2 | |
| Caltech101 | 84.8 | 49.2 | 93.6 | 87.6 | 76.1 | 93.8 | 97.4 | 70.6 | 91.6 | 34.0 | |
| EuroSAT | 84.8 | 49.2 | 93.6 | 87.6 | 76.6 | 93.9 | 97.4 | 70.7 | 91.7 | 87.4 | 83.3 |
| Average | 84.7 | 46.2 | 92.2 | 86.4 | 70.0 | 68.0 | 77.4 | 47.2 | 75.9 | 41.2 | 68.9 |

*Table 10.* Accuracy of Ours on the X-TAIL full-shot with order-II. Each row represents the performance on every dataset of the model trained after the corresponding task. Transfer , Average , and Last metrics are shown.

| | Cars | Aircraft | Pets | Food | SUN397 | MNIST | Flowers | DTD | Caltech101 | EuroSAT | |
|---|---|---|---|---|---|---|---|---|---|---|---|
| Transfer | | 23.8 | 87.8 | 84.3 | 61.4 | 41.6 | 65.7 | 34.5 | 71.0 | 28.9 | 55.4 |
| Cars | 86.9 | 23.8 | 87.8 | 84.3 | 61.2 | 43.9 | 65.7 | 36.4 | 72.4 | 37.4 | |
| Aircraft | 86.9 | 51.0 | 87.8 | 84.3 | 61.5 | 43.9 | 65.7 | 36.4 | 73.4 | 37.4 | |
| Pets | 86.9 | 51.0 | 94.3 | 84.4 | 61.5 | 40.1 | 65.7 | 36.3 | 73.0 | 37.4 | |
| Food | 86.9 | 51.0 | 94.3 | 89.4 | 61.5 | 40.1 | 65.7 | 35.8 | 73.2 | 36.7 | |
| SUN397 | 87.0 | 51.4 | 94.4 | 89.5 | 77.0 | 39.8 | 65.7 | 32.7 | 69.1 | 22.4 | |
| MNIST | 87.0 | 51.4 | 94.4 | 89.5 | 77.0 | 78.2 | 65.7 | 32.8 | 69.1 | 22.4 | |
| Flowers | 87.0 | 51.5 | 94.4 | 89.5 | 77.1 | 88.8 | 98.7 | 31.0 | 69.3 | 22.4 | |
| DTD | 87.0 | 51.5 | 94.4 | 89.6 | 77.1 | 95.4 | 98.7 | 75.5 | 68.6 | 22.2 | |
| Caltech101 | 87.0 | 55.4 | 94.5 | 89.6 | 77.4 | 98.8 | 99.0 | 75.7 | 95.7 | 21.9 | |
| EuroSAT | 87.0 | 55.4 | 94.5 | 89.6 | 77.7 | 98.8 | 99.0 | 75.8 | 95.8 | 95.8 | 86.9 |
| Average | 87.0 | 49.3 | 93.1 | 88.0 | 70.9 | 66.8 | 79.0 | 46.8 | 76.0 | 35.6 | 69.2 |

