# OpenReview forum: "LADA: Scalable Label-Specific CLIP Adapter for Continual Learning"
_ICML.cc/2025/Conference — ICML 2025 poster_

### Official Review · Reviewer_oKy3 · 2025-03-08

**Overall Recommendation:** 3

**Summary:**

This paper presents LADA (Label-specific ADApter), an approach for continual learning with vision-language models like CLIP. LADA enhances scalability and performance by generating discriminative, label-specific features. Unlike existing methods that partition parameters across tasks, LADA appends lightweight memory units to the frozen CLIP image encoder, enabling task-agnostic knowledge aggregation while preventing catastrophic forgetting through feature distillation. LADA ensures efficient training without updating the frozen parameters of CLIP. Experiment results show that LADA achieves state-of-the-art performance in continual learning.

**Claims And Evidence:**

The authors claim that “Our method is scalable and efficient, adding only small sets of learnable memory units for novel tasks, without requiring gradient propagation to the frozen CLIP image encoder.” However, the evidence provided in Table 4 primarily focuses on ablation studies without direct comparisons to other methods. To better support this claim, the authors are encouraged to include efficiency metrics of existing methods for a more comprehensive evaluation.

**Essential References Not Discussed:**

1. The related work section should include a dedicated subsection discussing continual learning (CL) methods based on CLIP that utilize parameter-efficient fine-tuning (PEFT) techniques, such as prompting, adapters, and LoRA.

2. While many early CL methods using PEFT involve a selection process for determining the appropriate prompt or adapter for a given task sample, recent efforts aim to avoid this selection step. The authors should discuss related works such as CODA-P (CVPR’23), APG (ICCV’23), and EvoPrompt (AAAI’24).

 3. The paper lacks discussion and comparison with prior CL methods that leverage prototypes for pseudo-feature replay. Some methods, such as HiDe-Prompt (NeurIPS’23), employ a single prototype per class, while others, such as CPP [1], utilize multiple prototypes per class. A discussion on these approaches would enhance the contextualization of LADA within the broader CL landscape.

[1] Steering prototype with prompt-tuning for rehearsal-free continual learning. WACV 2024

**Experimental Designs Or Analyses:**

The experimental design is well-structured, and the analyses are generally strong. However, additional comparisons in ablation studies, particularly in terms of computational efficiency, would further support the paper’s claims.

**Methods And Evaluation Criteria:**

Yes.

**Other Comments Or Suggestions:**

Missing “:” at the end of the line above Eq. (9).

**Other Strengths And Weaknesses:**

Other stengths.

1. The paper is generally easy to follow.

2. The proposed method demonstrates superior performance in Cross-domain TaskAgnostic Incremental Learning setting.

Other weaknesses.

1. Some notations are not clearly introduced. For example, what does the $\mathbf{i}$ represent in Eq. (5)?

2. More explaination is needed for the statement “a linear weighting between the logits produced by LADA and the corresponding text features”. Clarifying this process is crucial for understanding how final predictions for seen classes are made.

**Questions For Authors:**

My primary concern is the lack of discussion on related works. Please refer to the section “Essential References Not Discussed” for further details.

**Relation To Broader Scientific Literature:**

This work eliminates the need for auxiliary prompt or adapter selection, offering a unified and scalable solution for continual learning scenarios.

**Theoretical Claims:**

There’s no theoretical claims.

---

> ### Author Rebuttal · Authors · 2025-03-31
>
> Dear Reviewer oKy3,
>
> Thank you for your detailed review. We address your concerns one by one in the followings.
>
> > **W1 Claims And Evidence:** the evidence provided in Table 4 focuses on ablation without comparisons to other methods. **Experimental Designs Or Analyses:**  computational efficiency would further support the paper’s claims.
>
> **A1** We conducte a computational efficiency experiments. We compare our method with other methods using a consistent input batch size of 64 in the full-shot setting. Time comparisons for MoE-Adapters and ours were conducted on a single NVIDIA 4090 GPU. The results are summarized in the table below:
>
> |Method|Train Params (M)|Max GPU Memory (GB)|Time (s/batch)|
> |-|-|-|-|
> |LWF|149.6|31.42|--|
> |ZSCL|149.6| 25.67|--|
> |MoE-Adapters|59.8|21.83|0.337|
> |Ours| **11.2**|**18.51**|**0.289**|
>
> > **W2 Essential References Not Discussed:**
> >
> > 1. The related work should discuss CL methods based on CLIP that utilize PEFT techniques.
> > 2. The authors should discuss related works such as CODA-P, APG, and EvoPrompt.
>
> **A2** We appreciate this suggestion and will add a dedicated subsection titled **"Continual Learning Methods Based on PEFT"** to the related work section of the revised manuscript:
>
> Parameter-efficient fine-tuning (PEFT) has been widely adopted in continual learning to enhance representation learning for specific tasks. Prompt-based methods, such as L2P [1], expand the prompt pool as new tasks are learned, relying on auxiliary losses. **CODA-Prompt** [2] propose a end to end prompt selection methods to increase plasticity. **APG** [3] introduces a prompt generator to reduce the gap between pretraining and future tasks, while **EvoPrompt** [4] proposes an adaptive and continuous prompting approach to alleviate issues of selection mismatches and limited prompt shareability. MoE-Adapters [5] insert a mixture of adapters into the image encoder, activating a few for each task. However, these methods typically involve an auxiliary parameter selection step during inference to extract image features using the expected prompts or adapters, which can lead to misassignments and degrade classification performance.
>
> > **W3 Essential References Not Discussed:** The paper lacks discussion and comparison with prior CL methods that leverage prototypes for pseudo-feature replay, such as HiDe-Prompt and CPP.
>
> **A3** We are grateful for the reviewer’s suggestion. We will incorporate a discussion on these approaches in the related work of the revised manuscript:
>
> Prototype-based methods leverage pseudo-feature replay to mitigate forgetting in continual learning. **HiDe-Prompt** [6] optimizes hierarchical components with task-specific prompts using a single prototype per class, while **Contrastive Prototypical Prompt** [7] employs task-specific prompt-tuning with contrastive learning and multiple prototypes per class to address semantic drift and prototype interference. However, as highlighted in our introduction, while backward forgetting has been effectively addressed, these approaches often overlook the impact of pseudo-feature replay on the model’s original generalization ability, leading to forward forgetting.
>
> > **W4** what does the **i** represent in Eq. (5)?
>
> **A4** We apologize for the oversight. The notation **i** represents the image embedding, as defined in line 115 of the original manuscript. We will explicitly clarify this directly at Eq. (5) in the revised version.
>
> > **W5** More explanation is needed for the statement “a linear weighting between the logits produced by LADA and the corresponding text features”. Clarifying this process is crucial for understanding how final predictions for seen classes are made.
>
> **A5** We regret the confusion. The linear weighting is a fixed hyperparameter (set to 1.0 in experiments), not learned, balancing LADA logits and text features equally. This will be clarified in the revision.
>
> > **W6 Questions For Authors:** My primary concern is the lack of discussion on related works.
>
> **A6**  We have rewritten the related work in **A2** and **A3** . We commit to integrating these revisions into the manuscript.
>
> [1] Wang, Zifeng, et al. Learning to prompt for continual learning. *CVPR*. 2022.
>
> [2] Smith, James Seale, et al. Coda-prompt: Continual decomposed attention-based prompting for rehearsal-free continual learning. *CVPR* 2023.
>
> [3] Tang, Yu-Ming, et al. When prompt-based incremental learning does not meet strong pretraining. *ICCV* 2023.
>
> [4] Kurniawan, Muhammad Rifki, et al. Evolving parameterized prompt memory for continual learning. *AAAI* 2024.
>
> [5] Yu, Jiazuo, et al. Boosting continual learning of vision-language models via mixture-of-experts adapters. *CVPR* 2024.
>
> [6] Wang, Liyuan, et al. Hierarchical decomposition of prompt-based continual learning: Rethinking obscured sub-optimality. *NeurIPS* 2023.
>
> [7]  Li, Zhuowei, et al. Steering prototypes with prompt-tuning for rehearsal-free continual learning. *WACV* 2024.

---

> > ### Comment · Reviewer_oKy3 · 2025-04-04
> >
> > Thanks for the rebuttal, which addresses most of my concerns, particularly regarding the clarification of technical details and the discussion of related work. Therefore, I maintain my original recommendation of "Weak accept".

---

> > > ### Author Response · Authors · 2025-04-04
> > >
> > > Thank you for your constructive feedback and for acknowledging our rebuttal. We appreciate your thoughtful review and are glad that our clarifications addressed your concerns.

---

### Official Review · Reviewer_vJJv · 2025-03-11

**Overall Recommendation:** 3

**Summary:**

This paper primarily proposes an adapter-based method initialized with class cluster centers for CLIP-based Cross-Domain Task-Agnostic Incremental Learning(X-TAIL). This approach enables class discrimination within a unified feature space using class-specific parameters without requiring task-specific parameter selection.
Additionally, it models old class features using the Gaussian Mixture Model (GMM) and gets old class features by sampling from the GMM distributions.
The approach achieves state-of-the-art performance in X-TAIL.

##update after rebuttal
After carefully reviewing the authors' responses and the other reviewers' comments, I appreciate that my main concerns have been adequately addressed. I am now inclined to recommend acceptance.

**Claims And Evidence:**

Yes, their claims are clear and convincing

**Essential References Not Discussed:**

There are several essential references related to Distribution-Preserved Training that are either briefly mentioned or not cited in the paper.
[1] proposes a method that augments prototypes using Gaussian noise. While it is mentioned in the paper, its similarity to the proposed approach, where GMM sampling is used to augment prototypes, warrants a more detailed discussion. [2], [3], and [4] all adopt the strategy of sampling old class features from a class-specific Gaussian distribution, yet they are not cited. The key distinction of this work is the use of a Gaussian Mixture Model (GMM) instead of a simple Gaussian distribution, which should be explicitly discussed.

Additionally, the method [5], which is highly relevant to the Training of Label-Specific CLIP Adapter, is only cited but not discussed. Although [5] focuses on a CLIP-based few-shot adapter, its overall structure, initialization strategy, training approach, non-linearity design, and final feature extraction process share significant similarities with the proposed method. A more detailed discussion is needed to clarify these connections.

[1] Zhu, F., Zhang, X. Y., Wang, C., Yin, F., & Liu, C. L. (2021). Prototype augmentation and self-supervision for incremental learning. In Proceedings of the IEEE/CVF conference on computer vision and pattern recognition (pp. 5871-5880).

[2] Tang, Y. M., Peng, Y. X., & Zheng, W. S. (2023). When prompt-based incremental learning does not meet strong pretraining. In Proceedings of the IEEE/CVF International Conference on Computer Vision (pp. 1706-1716).

[3] Zhang, G., Wang, L., Kang, G., Chen, L., & Wei, Y. (2023). Slca: Slow learner with classifier alignment for continual learning on a pre-trained model. In Proceedings of the IEEE/CVF International Conference on Computer Vision (pp. 19148-19158).

[4] Huang, L., Cao, X., Lu, H., & Liu, X. (2024, September). Class-incremental learning with clip: Adaptive representation adjustment and parameter fusion. In European Conference on Computer Vision (pp. 214-231). Cham: Springer Nature Switzerland.

[5] Zhang, R., Zhang, W., Fang, R., Gao, P., Li, K., Dai, J., ... & Li, H. (2022, October). Tip-adapter: Training-free adaption of clip for few-shot classification. In European conference on computer vision (pp. 493-510). Cham: Springer Nature Switzerland.

**Experimental Designs Or Analyses:**

The experimental design and analysis follow the setup established in previous works introducing this problem. The study utilizes ten different domain datasets as ten sequential tasks for continual learning. This experimental design is well-founded and appropriate.

**Methods And Evaluation Criteria:**

The proposed method makes sense for the problem

**Other Comments Or Suggestions:**

In the ablation study results, the improvement of BF achieved by the presented modules appears to be limited. Further analysis is needed to clarify and justify the effectiveness of these modules.
Additionally, on page 5, in the "Overall Framework" section, the authors mention that "the final prediction is obtained by applying a linear weighting between the logits produced by LADA and the corresponding text features." It is unclear whether this linear weighting is a hyperparameter and how it is determined. Further clarification on this aspect would be beneficial.

**Other Strengths And Weaknesses:**

The novelty of this paper is questionable. The key module, Label-Specific CLIP Adapter, bears a strong resemblance to the CLIP-based few-shot learning method Tip-adapter [5]. Tip-adapter initializes the adapter by using multiple features from same-class images to set the corresponding class-specific weights in the adapter. It also explores fine-tuning this initialized adapter to further improve performance. The adapter structures are highly similar, and the output formulation of Tip-adapter is identical to Equation (6) in this paper. Additionally, both methods employ the same technique to convert inner products into non-negative values. The overall framework follows a similar strategy, where the final classification result is obtained by weighting the adapter’s output and the text encoder’s output.
The primary difference lies in the experimental setup: this paper focuses on full-shot datasets, allowing the use of K-means to obtain class sub-centers for initialization, whereas Tip-adapter, designed for few-shot scenarios, directly initializes the adapter using a limited number of sample features. Additionally, this paper incorporates a continual learning setting, introducing the freezing of old class classifiers—a widely used technique in the continual learning domain. However, this adaptation is a general strategy rather than a novel methodological contribution.

**Questions For Authors:**

Similarity to Tip-adapter: The proposed Label-Specific CLIP Adapter appears to be highly similar to Tip-adapter, with key similarities in initialization, structure, and final classification formulation. The primary differences seem to be (1) using K-means clustering for adapter initialization instead of directly leveraging a few-shot feature set, and (2) incorporating a continual learning setting with frozen classifiers for old classes. Given these similarities, could the authors clarify the novelty of their approach beyond these modifications?

**Relation To Broader Scientific Literature:**

This work builds upon prior research in incremental learning and pre-trained model fine-tuning. Unlike prompt tuning in incremental learning areas, there is no need to select additional parameters. Class features are trained using kmeans cluster center initialization instead of random initialization of the vanilla adapter

**Theoretical Claims:**

The paper does not include formal theoretical Claims or proofs.

---

> ### Author Rebuttal · Authors · 2025-03-31
>
> Dear Reviewer vJJv,
>
> Thank you for your detailed review. We address your concerns one by one in the followings.
>
> > **W1 Other Strengths And Weaknesses:** Label-Specific CLIP Adapter bears a strong resemblance to the CLIP-based few-shot learning method Tip-Adapter [5]. **Questions For Authors:** Similarity to Tip-Adapter.
>
> **A1** We agree with the reviewer that the adapter module in LADA shares similarities with Tip-Adapter. **However, our key contribution lies in generalizing this module for continual learning tasks—a scenario where Tip-Adapter fails to perform effectively.**
>
> We demonstrate that the adapter (which we term *label-specific memory*) can naturally capture task-specific data distributions and mitigate forgetting. To maintain scalability and simplicity, we initialize the memory cache using k-means clustering and adopt Tip-Adapter’s output formulation. In our view, adapting Tip-Adapter to continual learning is nontrivial: LADA introduces significant improvements to address both backward and forward forgetting.
>
> Moreover, LADA seamlessly extends to few-shot learning, matching Tip-Adapter’s original setting. Under the 16-shot benchmark (using CLIP ViT-B/16, following [6]), LADA outperforms Tip-Adapter-F by **1.6%** (see table below). Notably, it achieves this with only **1/4** of the feature embeddings required by Tip-Adapter, underscoring its efficiency.
>
> ||Aircraft|Caltech101|DTD|EuroSAT|Flowers|Food|Pets|Cars|Sun397|Average|
> |-|:-:|:-:|:-:|:-:|:-:|:-:|:-:|:-:|:-:|:-:|
> |Tip-Adapter-F|44.6|95.7|70.8|85.9|96.2|86.8|92.6|82.3|76.0|81.2|
> |Ours|**49.6**|**96.4**|**71.5**|**86.8**|**97.4**|**87.7**|**94.0**|**84.8**|**76.8**|**82.8**|
>
> The performance results for Tip-Adapter-F are sourced from recent work [6].
>
> **Summary of  contributions**
>
> **Contribution 1:** We design a text encoder fine-tuning framework which enhances the classification ability and naturally obtains the transfer ability of CLIP model.
>
> **Contribution 2:** We propose LADA, a lightweight CLIP adapter that condenses task-agnostic knowledge to transform CLIP representation into label-specific features, eliminating the need for parameter selection as seen in previous mainstream methods.
>
> **Contribution 3:** We propose Distribution-Preserved Training which controls the influence of prototypes by incorporating their contribution weights into the loss function. We not only addresses backward forgetting but also tackles forward forgetting, a challenge overlooked in prior methods.
>
> > **W2 Essential References Not Discussed:** There are several essential references related to Distribution-Preserved Training that are either briefly mentioned or not cited in the paper.
>
> **A2** Thanks for the reviewer highlighting these relevant papers and we will include citations and discussions. Our approach aligns with a standard Gaussian distribution when $\lambda_2=1$, similar to the prototype augmentation method in [1]. When $\lambda_2 > 1$, our loss function (Equation 10) leverages the mixture weights $\pi$ from the GMM to perform more sophisticated augmentation. As shown in Table 4, when $\lambda_2=1$, the *Transfer* performance is lower compared to when $\lambda_2 > 1$. This indicates that the effectiveness of DPT while prior methods in [1-4] do not address *forward forgetting* in pre-trained models, our approach explicitly tackles this issue.
>
> > **W3 Other Comments Or Suggestions:** The improvement of BF achieved by the presented modules appears to be limited.
>
> **A3** In Tab.2 and Tab.3, our basic fine-tuning framework (BF) already outperforms several baseline methods, demonstrating the challenges prior approaches face in task-agnostic settings. In the full-shot setting, these modules improve *Transfer* by an average of **2.3%**, *Average* by **2.0%**, and *Last* by **1.3%** compared to the BF framework alone.
>
> > **W4 Other Comments Or Suggestions:** It is unclear whether this linear weighting is a hyperparameter.
>
> **A4** We regret the confusion. The linear weighting is a fixed hyperparameter (set to 1.0 in experiments), not learned, balancing LADA logits and text features equally. This will be clarified in the revision.
>
> We hope this rebuttal addresses your concerns. Please let us know if further refinements are needed!
>
> [1] Zhu, Fei, et al. Prototype augmentation and self-supervision for incremental learning. *CVPR* 2021.
>
> [2] Tang, Yu-Ming, et al. When prompt-based incremental learning does not meet strong pretraining. *ICCV* 2023
>
> [3] Zhang, Gengwei, et al. Slca: Slow learner with classifier alignment for continual learning on a pre-trained model. *ICCV* 2023.
>
> [4] Huang, Linlan, et al. Class-incremental learning with clip: Adaptive representation adjustment and parameter fusion. *ECCV* 2024.
>
> [5] Zhang, Renrui, et al. Tip-adapter: Training-free adaption of clip for few-shot classification. *ECCV* 2022.
>
> [6] Zanella, Maxime, et al. Low-rank few-shot adaptation of vision-language models. *CVPR* 2024.

---

### Official Review · Reviewer_EaVq · 2025-03-11

**Overall Recommendation:** 3

**Summary:**

Instead of partitioning parameters across tasks, this paper proposed LADA appended lightweight, labelspecific memory units to the frozen CLIP image encoder, enabling discriminative feature generation by aggregating task-agnostic knowledge. The method achieves state-of-the-art performance in continual learning settings on several datasets.

**Claims And Evidence:**

Yes

**Essential References Not Discussed:**

In [1], RANPAC also projects features into a higher-dimensional feature space, which has been proven to be beneficial for continual learning. It has not been demonstrated that the effect of LADA is better than that of nonlinear random projection [1].

[1]McDonnell, Mark D., et al. "Ranpac: Random projections and pre-trained models for continual learning." NeurIPS,2023.

**Experimental Designs Or Analyses:**

Most experiments were checked. Please refer to Other Strengths and Weaknesses.

**Methods And Evaluation Criteria:**

It makes sense in general

**Other Comments Or Suggestions:**

Please refer to Other Strengths and Weaknesses.

**Other Strengths And Weaknesses:**

1. It is recommended to add a flowchart of the LADA method or end-to-end loss function in the Overall Framework section, which will make the method better understood.
2. LADA projects features into a higher-dimensional feature space, which has been proven to be beneficial for continual learning [1]. However, in the ablation experiment, it has not been demonstrated that the effect of LADA is better than that of nonlinear random projection [1]. I think [1] should be included in the reference.
3. Experiments on some important datasets are necessary, such as CIFAR100. It is recommended to supplement.

[1]McDonnell, Mark D., et al. "Ranpac: Random projections and pre-trained models for continual learning." NeurIPS,2023.

**Questions For Authors:**

Please refer to Other Strengths and Weaknesses.

**Relation To Broader Scientific Literature:**

The method appends lightweight, label-specific memory units to the frozen CLIP image encoder, enabling discriminative feature generation by aggregating task-agnostic knowledge.

**Theoretical Claims:**

Most proofs were checked.

---

> ### Author Rebuttal · Authors · 2025-03-31
>
> Dear Reviewer EaVq,
>
> Thank you for your detailed review. We address your concerns one by one in the followings.
>
> > **W1**: **Essential references not discussed of [1]** and the second point in Other Strengths And Weaknesses of **comparisons with RanPAC [1]**
>
> **A1**: We acknowledge the reviewer’s comments regarding the article and address the concerns about the omission of RanPAC [1] as an essential reference, as well as the need for a comparison between RanPAC and our proposed method.
>
> 1. Core Idea of RanPAC
>
>    RanPAC [1] projects features into a higher-dimensional space, which has been shown to benefit continual learning. RanPAC [1] tackles multicollinearity in CLIP’s original feature dimensions by employing **random Gaussian projection matrices** and **nonlinear activation functions** (e.g., ReLU). Specifically, it extracts CLIP features using a frozen encoder, projects them via a Gaussian matrix $W$, applies nonlinear transformations, and uses ridge regression on 0-1 encoded labels to train classifiers. This approach leverages second-order statistics to reduce feature redundancy and enhance classifier accuracy.
>
> 2. Key Differences and Advantages of LADA
>
>    - **Feature Optimization**
>
>      RanPAC optimizes features using static **random Gaussian projection matrices** to reduce multicollinearity in CLIP features, but it lacks adaptability to task-specific nuances, often failing to capture fine-grained or diverse class characteristics effectively. In contrast, LADA introduces a **label specific memory** dynamically amplifying CLIP feature most relevant to each class. This process condenses features into sparse, high-dimensional representations tailored to individual tasks, enhancing discriminability without modifying the frozen CLIP encoder. By leveraging this class-specific adaptability, LADA achieves superior plasticity and stability.
>
>    - **Cross-Task Adaptation**
>
>      RanPAC’s dependence on **fine-tuning the CLIP encoder for the first task** incurs substantial computational overhead and risks eroding CLIP’s pre-trained general knowledge, making it impractical for continual learning across diverse tasks. In contrast, LADA takes a fundamentally different approach by **freezing the CLIP encoder entirely** and training lightweight a **label-specific adapter**.
>
> 3. Empirical Validation
>
>    To compare LADA with RanPAC, we integrate RanPAC into our framework (denoted as BF+RanPAC+DPT, where BF refers to baseline fine-tuning and DPT to Distribution-Preserved Training) and evaluate it against our method (BF+LADA+DPT). In the X-TAIL 16-shot setting, the results demonstrate that BF+LADA+DPT outperforms BF+RanPAC+DPT by **1.4% in Transfer**, **1.4% in Average**, and **1.0% in Last** metrics, averaged across the 10 tasks.
>
> ||Aircraft|Caltech101|DTD|EuroSAT|Flowers|Food101|Mnist|Pets|Cars|Sun397   |***Average***|
> |-|:-:|:-:|:-:|:-:|:-:|:-:|:-:|:-:|:-:|:-:|:-:|
> | **Transfer**|
> |BF+**RanPAC**+DPT|--|74.4|35.4|34.6|63.4|83.3|36.9|87.6|64.7|60.8|60.1|
> |Ours|--|**75.0**|**36.1**|**35.9**|**66.3**|**83.7**|**42.1**|**88.0**|**65.3**|**61.4**|**61.5**|
> |**Average**|
> |BF+**RanPAC**+DPT|47.0|86.7|61.0|71.4|83.3|84.3|59.5|89.4|68.6|62.2|71.3|
> |Ours|**49.1**|**91.0**|**61.3**|**71.6**|**84.4**|**85.0**|**62.8**|**89.7**|**69.2**|**62.9**|**72.7**|
> |**Last**|
> |BF+**RanPAC**+DPT|47.1|90.3|68.3|87.2|96.5|85.4|93.4|**93.7**|84.2|74.4 |82.1|
> |Ours|**49.6**|**93.7**|**69.3**|**86.9**|**96.7**|**86.9**|**93.8**|**93.7**|**84.6**|**76.0**|**83.1**|
>
> [1] McDonnell, Mark D., et al. "Ranpac: Random projections and pre-trained models for continual learning." NeurIPS,2023.
>
> > **W2**: It is recommended to add a flowchart of the LADA method or end-to-end loss function in the Overall Framework section
>
> **A2**: We appreciate the reviewer’s suggestion. The overall framework and end-to-end loss function of LADA are available in an [anonymized  link](https://anonymous.4open.science/r/ICML25-LADA-4B78/README.md). We will include them in the revised manuscript.
>
> > **W3**: Experiments on some important datasets are necessary, such as CIFAR100.
>
> **A3**: Thank you for your suggestion. CIFAR-100 is indeed an important dataset. However, due to overlapping class names with other datasets, it was not included in the original X-TAIL dataset to ensure accurate evaluation. Nevertheless, we have conducted additional experiments on **Inclusion of CIFAR-100 in the X-TAIL** to further demonstrate the robustness of our approach. Under the 16-shot setting, our method still achieves state-of-the-art results, with improvements of **4.3%**, **2.0%**, and **1.0%** on the Transfer, Average, and Last metrics, respectively. Please refer to the [anonymized  link](https://anonymous.4open.science/r/ICML25-LADA-4B78/README.md) for detailed experimental results.
>
> We hope this rebuttal addresses your concerns. Please let us know if further refinements are needed!

---

> > ### Comment · Reviewer_EaVq · 2025-04-05
> >
> > Thank you for the responses, which have addressed most of my concerns. I have re-rated the paper to “weak accept”.

---

> > > ### Author Response · Authors · 2025-04-05
> > >
> > > Thank you sincerely for your thoughtful review and for carefully considering our responses. We truly appreciate your recognition of our efforts in addressing the concerns, and we are grateful for your updated rating.

---

### Official Review · Reviewer_4dD8 · 2025-03-15

**Overall Recommendation:** 3

**Summary:**

The paper proposes a task incremental learning approach for CLIP encoders. In contrast to using adapters for the image encoder, it learns task (or class) specific memories represented by learnable vectors. The classification is performed via dot products of these task specific memories and CLIP image embeddings. The training mechanism attempts to prevent forgetting by multiple techniques including freezing memories of old tasks, separating samples from old tasks from the new ones, suitable modeling of distribution of clusters of past memories etc. The method also performs fine tuning for the text encoder for the new classes with mechanism to prevent forgetting.

The method was tested on multiple standard datasets, compared against relevant baselines and evaluated in different setting to illustrate capability of learning new classes as well as retaining classification performances on the old ones.

**Claims And Evidence:**

---

**Essential References Not Discussed:**

---

**Experimental Designs Or Analyses:**

Sufficient experimentation (with analysis) was performed to show the strength of the method.  Evaluation in 3 settings (transfer, last and avg) as well as illustration of performance consistency over learning steps in Fig 2 are both good attempts to convince the reader of the merits of this approach.
An analysis or ablation on why GMM modeling was needed would be good to justify this choice.

**Methods And Evaluation Criteria:**

I found the method distribution is sloppy and confusing.

1. It is not mentioned in the introduction or at the beginning of the method section that the proposed method also finetunes the text encoder. Line 121 starts a new section titled "A simple baseline" which confused me as whether this is a baseline method that was compared against or part of the overall approach -- I think it is the latter. It should have been clearly mentioned that finetuning text encoder is part of the proposed approach.

2. The finetuning of text encoder does not clearly mention the full form of the loos function used. If I understood correctly, Eqns 3 and 4 are used for current and past tasks respectively and the total loss is a summation of the two. But it is not clear in the text.

3. Similarly, Line 206 suddenly, and without much justification, introduces GMM modeling of the p^i_j vectors. This again would confuse the reader, is Eqn 8 or 10 used for previous task?  Here too, the writeup does not clearly state the full loss function.

4. The introduction is unnecessarily long and not well written.

**Other Comments Or Suggestions:**

Is "CLIP Adapter" the right term for the technique proposed? Would it not give the impression that the method is using adapters for image encoder before one reads the intro?

Something like Task Memories for Scalable Continual Learning with CLIP Encoders might be more reflective of the method? Just a thought.

**Other Strengths And Weaknesses:**

---

**Questions For Authors:**

---

**Relation To Broader Scientific Literature:**

Will rely on other reviewers for novelty assessment.

**Theoretical Claims:**

---

---

> ### Author Rebuttal · Authors · 2025-03-31
>
> Dear Reviewer 4dD8,
>
> Thank you for your detailed review. We address your concerns one by one in the followings.
>
>
>
> > **W1** It should have been clearly mentioned that finetuning text encoder is part of the proposed approach.
>
> **A1** We sincerely apologize for the lack of clarity and fully agree with your assessment that the role of text encoder finetuning should have been explicitly stated early in the paper. We will make the following modifications in the next version.
>
> 1. **Introduction (Section 1):**
>
>    - We will add a paragraph in the introduction explicitly stating that our proposed approach jointly optimizes both the image features and text encoders. Specifically, we will emphasize that we first establish a text encoder fine-tuning approach for continual learning by combining parameter freezing and distillation techniques.
>
> 2. **Methodology (Section 3):**
>
>    - We will explicitly outline the full pipeline, including text encoder fine-tuning for continual learning, before introducing implementation details.  Also, an [overall framework](https://anonymous.4open.science/r/ICML25-LADA-4B78/README.md) will be added, illustrating both text encoder fine-tuning module and Label Specific CLIP Adapter module.
>    - The section titled **"A Simple Baseline via Text Encoder Fine-tuning"**  will be renamed to **"Text Encoder Fine-tuning Framework for Continual Learning"** to clarify its role as an ablated variant of our approach. We will add a transition sentence:  "To address continual learning in X-TAIL senario, we first introduce a baseline framework with text encoder finetuning only."
>
>
>
> > **W2** The finetuning of text encoder does not clearly mention the full form of the loss function used. If I understood correctly, Eq. 3 and 4 are used for current and past tasks respectively and the total loss is a summation of the two.
>
> **A2** We apologize for the confusion. In the revised manuscript, we will add the following sentence in Section 3.2 after Equation 4:  **"The total training loss is the sum of the current task loss (Eq. 3) and the distillation loss (Eq. 4), enabling joint optimization of both current and past task objectives."**
>
>
>
>
> > **W3** Is Eq. 8 or 10 used for previous task? Here too, the writeup does not clearly state the full loss function.
>
> **A3** We will clarify that  total loss combines Eq. 7 (current task) and **Eq. 10** (Distribution-Preserved Training loss for **previous task**) in next version.
>
>
>
> > **W4** The introduction is unnecessarily long and not well written.
>
> **A4** Thank you for the constructive feedback. We will revise the introduction by:
>
> 1. Merge paragraphs 2-3 to concisely present the dual challenges of *memory stability* (preserving past knowledge) and *learning plasticity* (adapting to new tasks) in continual learning.
> 2. Remove detailed discussions of prior methods (e.g., regularization-based approaches) and shift them to the Related Work section.
>
>
>
> > **W5** An analysis or ablation on why GMM modeling was needed would be good to justify this choice.
>
> **A5** We acknowledge that the the choice of GMM should be explicitly stated. We will add the following discussions in the next version of paper.
>
> **Choice of GMM**: Distilling only cluster centers loses fine-grained distribution information. GMM models the **full feature distribution** of past tasks, which controls the influence of prototypes by incorporating their contribution weights $\pi$ into the loss function Eq. 10.
>
> **Impact of GMM**: Table 3 shows DPT based on GMM boosts Transfer and Average performance.  Table 4 shows increasing the number of image prototypes $\lambda_2$ further improves Transfer performance.
>
>
>
> > **W6** Is "CLIP Adapter" the right term for the technique proposed? Would it not give the impression that the method is using adapters for image encoder before one reads the intro?
>
> **A6** Thank you for raising this concern. The term "CLIP-Adapter" has been recognized in vision-language research [1] to denote lightweight feature adaptation after the frozen CLIP backbone extracts representations, not input-level modifications. This naming convention emphasizes post-encoder refinement for downstream tasks, aligning with its technical definition and community usage. We will clarify this distinction explicitly in the revised manuscript to avoid potential misinterpretation.
>
> [1] Gao, Peng, et al. Clip-adapter: Better vision-language models with feature adapters. *International Journal of Computer Vision* 132.2 (2024): 581-595.
>
>
>
> > **W7** Will rely on other reviewers for novelty assessment.
>
> **A7** Please refer to **A1** and **Summary of  contributions** of **reviewer vJJv**.
>
>
>
> We hope this rebuttal addresses your concerns. Please let us know if further refinements are needed!

---

### Decision · Program_Chairs · 2025-05-01

**Decision:**

Accept (poster)

**Comment:**

This paper proposes LADA (Label-specific ADApter), a method for continual learning with CLIP-based models that avoids task-specific parameter partitioning. Instead, LADA appends lightweight, label-specific memory units to the frozen CLIP image encoder, enabling discriminative feature generation while preventing catastrophic forgetting through Distribution-Preserved Training (DPT). The method achieves state-of-the-art performance in Cross-domain Task-Agnostic Incremental Learning (X-TAIL) by combining task-agnostic knowledge aggregation with efficient training (no updates to CLIP’s frozen parameters).

LADA presents a practically effective solution for CLIP-based continual learning, with strong empirical results and scalability advantages. While the initial writing needed refinement, the rebuttal convincingly addressed critiques. The paper’s real-world applicability (minimal overhead, no CLIP fine-tuning) and unified approach (no task-specific selection) align with ICML’s emphasis on scalable ML. Post-rebuttal revisions solidify its contribution.